# Learning to Add, Multiply, and Execute Algorithmic Instructions Exactly with Neural Networks

**Artur Back de Luca**[*]
University of Waterloo
abackdel@uwaterloo.ca

**George Giapitzakis**[*]
University of Waterloo
ggiapitz@uwaterloo.ca

**Kimon Fountoulakis**
University of Waterloo
kfountou@uwaterloo.ca

## Abstract

Neural networks are known for their ability to approximate smooth functions, yet they fail to generalize perfectly to unseen inputs when trained on discrete operations. Such operations lie at the heart of algorithmic tasks such as arithmetic, which is often used as a test bed for algorithmic execution in neural networks. In this work, we ask: can neural networks learn to execute binary-encoded algorithmic instructions exactly? We use the Neural Tangent Kernel (NTK) framework to study the training dynamics of two-layer fully connected networks in the infinite-width limit and show how a sufficiently large ensemble of such models can be trained to execute exactly, with high probability, four fundamental tasks: binary permutations, binary addition, binary multiplication, and Subtract and Branch if Negative (SBN) instructions. Since SBN is Turing-complete, our framework extends to computable functions. We show how this can be efficiently achieved using only logarithmically many training data. Our approach relies on two techniques: structuring the training data to isolate bit-level rules, and controlling correlations in the NTK regime to align model predictions with the target algorithmic executions.

## 1 Introduction

There has been growing interest in the computational capabilities and efficiency of neural networks both from a theoretical and empirical perspective [7, 16, 21, 28, 31, 34]. Most works have either demonstrated the expressive power of different architectures through simulation results or learnability from a probably approximately correct (PAC) viewpoint. While important, the mere existence of parameter configurations that realize a specific computation or a generalization bound does not offer insight into the ability to learn to execute an algorithm through gradient-based training. In fact, simulating algorithmic instructions with neural networks often involves approximating discontinuous functions that gradient descent is difficult to converge to with standard training datasets [1].

By modeling training with the Neural Tangent Kernel (NTK), we prove that two-layer fully connected networks in the infinite-width limit can learn to iteratively execute *binary permutations*, *binary addition*, *binary multiplication*, and *SBN instructions* with a logarithmic number of examples, or equivalently, a number of examples polynomial in the input bit size. Rather than training on traditional input-output pairs, we exploit the locality of these algorithms by casting each step as a set of templates. We show that training with these local templates is sufficient for full algorithmic execution when composed across iterations of a loop. To our knowledge, this is the first NTK-based proof of exact learnability for these tasks. A high-level overview of our approach is shown in Figure 1.

**Contributions.** Our approach is built on two key innovations that overcome the interference and ambiguity in training data that can usually create problems for neural learning for discrete tasks:

1. **Algorithmic template representation:** We demonstrate how to design training data that represent local computations (i.e., operating only on a subset of bits) that can be composed

39th Conference on Neural Information Processing Systems (NeurIPS 2025).

---

[*]Equal contribution

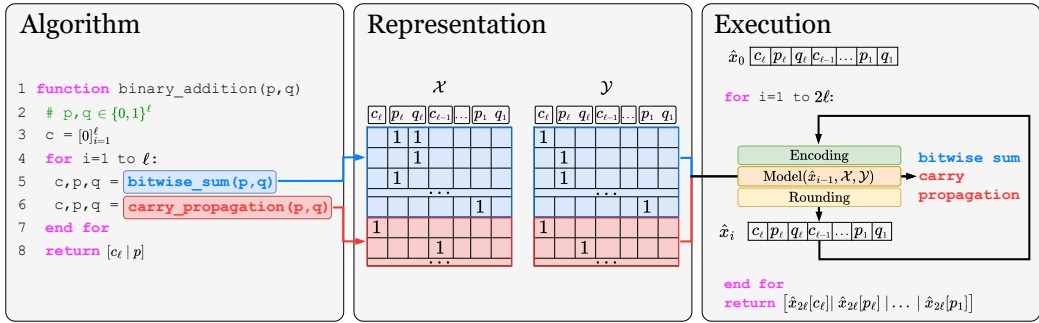

Figure 1: Simplified illustration of the framework used in our analysis. The left panel shows an example algorithm (binary addition) where each function, highlighted in blue and red, is translated into binary training instructions shown in the central panel with matching colors. Each instruction specifies a condition over part of the current algorithm state and maps it to a corresponding output. Instructions are grouped into blocks, indicated by boxed column labels in $\mathcal{X}$ and $\mathcal{Y}$, each representing a subset of the input state. For binary addition, some blocks represent segments of the summands, while others reflect the carry state. For the applications discussed in Section 6, this block structure allows the number of instructions in $\mathcal{X}$ and $\mathcal{Y}$ to scale linearly with bit length $\ell$. The right-most panel shows how instructions are used within an iterative framework to update the state vector $\hat{x}_i$, which serves as input to the neural network at the $i$-th step. The state is first encoded, as described in Section 5, before being passed to the model. In the NTK regime, we show that the network performs template matching against training samples to execute the appropriate instructions. As $\hat{x}_i$ evolves, it activates new templates, progressing through the algorithm. Predictions are rounded at each step to mitigate noise, and repeating this process reproduces the algorithm's full execution.

> to execute complete algorithms. Each algorithmic instruction is represented by "templates" and entire algorithms (e.g., binary permutations, binary addition, binary multiplication, and SBN instruction execution) can be executed by iteratively matching these templates. The total number of templates is logarithmic in the number of all possible inputs.
>
> 2. **Provable exact learnability:** We prove that, by training on an orthonormalized version of our templates, we can control unwanted correlations in the NTK regime and show that an ensemble of two-layer fully connected networks in the infinite-width limit can learn to execute algorithmic instructions exactly with high probability.

## 2   Literature review

The NTK framework [14] provides insight into continuous-time gradient descent (gradient flow) in fully connected feed-forward neural networks. It has been extended to discrete gradient descent [18] and generalized to other architectures like recurrent neural networks (RNNs) and Transformers [39, 40]. Although NTK theory has been widely developed, few works offer task-specific guarantees. Notable examples include [3], which proves that Transformers can generalize to unseen symbols for a class of pattern-matching language tasks. Our work addresses a different setting: we show that shallow feed-forward neural networks in the infinite-width limit trained by gradient descent can learn to exactly execute algorithmic instructions using a logarithmic number of examples.

Complementing NTK-based results, other studies have demonstrated the expressive power of neural architectures by simulating algorithmic tasks. Following this approach, [34] proves that RNNs are Turing-complete, and [11] demonstrates that RNNs can solve the shortest path problem and approximate solutions to the knapsack problem. Similarly, simulation results on Transformers also establish Turing completeness [28] and present constructive solutions that generalize across input lengths for arithmetic tasks [4], linear algebra [7, 41], graph-related problems [1], and parallel computation [2]. From a learnability perspective, [37] provides statistical guarantees for learning Turing-computable functions, while [20] shows that predictors trained auto-regressively can approximate any such functions. However, these approaches either depend on hand-crafted parameter configurations without

training or build upon the PAC framework, and therefore do not address the core question of exact learnability explored in this work.

Learning to execute algorithms has been the focus of numerous empirical studies. These works explore architectural modifications and prompting techniques, particularly aimed at improving generalization in arithmetic tasks [5, 6, 15, 21, 22, 23, 24, 27, 29, 31, 36, 38, 42]. In this context, [16] introduced the Neural GPU, which learns binary addition and multiplication and generalizes to sequences longer than those seen in training. Similarly, [35] proposed the Neural Arithmetic Logic Unit (NALU), which incorporates arithmetic operations into network modules to improve generalization. This approach was further refined in [19] with the introduction of the Neural Addition Unit (NAU) and Neural Multiplication Unit (NMU), offering better stability and convergence. More recent work aims to enhance the length generalization capabilities of Transformers through improved Positional Encodings [15, 30, 32, 43], the use of scratchpads [26], and prompting techniques for large language models (LLMs), such as Chain-of-Thought (CoT) programming [5, 6]. These empirical efforts provide practical methods to improve both in-distribution and out-of-distribution performance of neural networks on arithmetic tasks. However, they lack formal guarantees regarding the conditions under which generalization occurs. Our theoretical analysis offers precise sufficient criteria under which a neural network in the infinite-width limit can provably learn to execute algorithmic instructions.

## 3 Notation and preliminaries

Throughout the text, we use boldface to denote vectors. The symbols $\mathbf{1}$ and $\mathbf{0}$ denote the vector of all ones and zeros of appropriate length, respectively. We use the notation $[n]$ to refer to the set $\{1, 2, \ldots, n\}$. For a vector $\boldsymbol{x} \in \mathbb{R}^n$ we denote by $\|\boldsymbol{x}\| := \sqrt{\sum_{i=1}^n x_i^2}$ the Euclidean norm of $\boldsymbol{x}$. We denote the $n \times n$ identity matrix by $I_n$.

**Model and NTK Results.** We provide an overview of the theory used to derive our results. We refer the reader to [9] for a comprehensive treatment of the NTK theory. We work with two-layer, ReLU-activated fully connected feed-forward neural networks with no bias. Concretely, the architecture is defined as the function $F : \mathbb{R}^{k'} \to \mathbb{R}^k$ with $F(\boldsymbol{x}) = W^2 \operatorname{ReLU}(W^1 \boldsymbol{x})$ where $W^1 \in \mathbb{R}^{n_h \times k'}$, $W_2 \in \mathbb{R}^{n_h \times k}$, and $n_h \in \mathbb{N}$ is the hidden dimension. The weights are initialized according to the NTK parametrization as $W_{ij}^1 = \frac{\sigma_\omega}{\sqrt{k'}} \omega_{ij}^1$ and $W_{ij}^2 = \frac{\sigma_\omega}{\sqrt{n_h}} \omega_{ij}^2$ where $\omega_{ij}^1$ and $\omega_{ij}^2$ are trainable parameters initialized i.i.d. from a standard Gaussian distribution. When $n_h \to \infty$, the empirical NTK kernel given by $\nabla_{\{W^1, W^2\}} F(\boldsymbol{x})^\top \nabla_{\{W^1, W^2\}} F(\boldsymbol{x}')$ converges to the deterministic limit:

$$\Theta(\boldsymbol{x}, \boldsymbol{x}') = \left( \frac{\boldsymbol{x}^\top \boldsymbol{x}'}{2\pi k'} (\pi - \theta) + \frac{\|\boldsymbol{x}\| \cdot \|\boldsymbol{x}'\|}{2\pi k'} ((\pi - \theta) \cos \theta + \sin \theta) \right) I_k \in \mathbb{R}^{k \times k} \tag{1}$$

and the NNGP kernel is given by

$$\mathcal{K}(\boldsymbol{x}, \boldsymbol{x}') = \left( \frac{\|\boldsymbol{x}\| \cdot \|\boldsymbol{x}'\|}{2\pi k'} ((\pi - \theta) \cos \theta + \sin \theta) \right) I_k \in \mathbb{R}^{k \times k}, \tag{2}$$

where $\theta = \arccos \left( \boldsymbol{x}^\top \boldsymbol{x}' / \|\boldsymbol{x}\| \cdot \|\boldsymbol{x}'\| \right)$. For a set of vectors $\mathcal{X}$, we will use the notation $\Theta(\mathcal{X}, \cdot)$, $\Theta(\cdot, \mathcal{X})$ and $\Theta(\mathcal{X}, \mathcal{X})$ to refer to the limit NTK calculated when the the set $\{F(\boldsymbol{x}) : \boldsymbol{x} \in \mathcal{X}\}$ is vectorized (the outputs are stacked vertically), and similarly for the NNGP kernel. Our learnability results rely on the following theorem by [18], adapted to our architecture:

**Theorem 3.1** (Theorem 2.2 from Lee et al. 18). *Let $\mathcal{X}$ and $\mathcal{Y}$ be the training dataset (training inputs and ground truth labels, respectively). Assume that $\Theta := \Theta(\mathcal{X}, \mathcal{X})$ is positive definite. Suppose the network is trained with gradient descent (with small-enough step-size) or gradient flow to minimize the empirical MSE loss.[2] Then, for every $\hat{\boldsymbol{x}} \in \mathbb{R}^{k'}$ with $\|\hat{\boldsymbol{x}}\| \leq 1$, as $n_h \to \infty$, the output at training time $t$, $F_t(\hat{\boldsymbol{x}})$, converges in distribution to a Gaussian with mean and variance given by*

$$\mu(\hat{\boldsymbol{x}}) = \Theta(\hat{\boldsymbol{x}}, \mathcal{X}) \Theta^{-1} \mathcal{Y} \tag{3}$$

$$\Sigma(\hat{\boldsymbol{x}}) = \mathcal{K}(\hat{\boldsymbol{x}}, \hat{\boldsymbol{x}}) + \Theta(\hat{\boldsymbol{x}}, \mathcal{X}) \Theta^{-1} \mathcal{K}(\mathcal{X}, \mathcal{X}) \Theta^{-1} \Theta(\mathcal{X}, \hat{\boldsymbol{x}}) - (\Theta(\hat{\boldsymbol{x}}, \mathcal{X}) \Theta^{-1} \mathcal{K}(\mathcal{X}, \hat{\boldsymbol{x}}) + h.c) \tag{4}$$

*where $\mathcal{Y}$ in Equation (3) denotes the vectorization of all vectors $\boldsymbol{y} \in \mathcal{Y}$, and "h.c." is an abbreviation for the Hermitian conjugate.*

---

[2]The empirical MSE loss is defined as $\mathcal{L}(\mathcal{D}) = (2|\mathcal{D}|)^{-1} \sum_{(\boldsymbol{x}, \boldsymbol{y}) \in \mathcal{D}} \|f_t(\boldsymbol{x}) - \boldsymbol{y}\|^2$.

# 4 Algorithmic execution via template matching

In this section, we introduce a template matching principle that offers a high-level intuition for representing and executing algorithmic instructions in the NTK regime. This principle transforms binary inputs into binary outputs by comparing input configurations against a set of predefined templates. Matched templates are then used to compose the corresponding output. We use this principle of templates and template matching functions to encode and execute algorithms. While this approach may seem unrelated to neural networks, we show in the following sections that, by carefully structuring training and test inputs, neural networks can emulate this template matching mechanism. In doing so, they can learn algorithms on binary data through the lens of the Neural Tangent Kernel. More concretely, we show that this principle, as described here, allows us to learn and express arithmetic operations such as addition and multiplication. Furthermore, this approach can be generalized to more general computations, as discussed in Section 5.

**Input representation:** Inputs are vectors of binary variables of length $k$, i.e., $\hat{\boldsymbol{x}} \in \{0, 1\}^k$. These variables are grouped into disjoint blocks, partitioning $\hat{\boldsymbol{x}}$ into $b$ blocks. Each block, denoted by $B_i$ for $i \in [b]$, has a length $s_i \in \mathbb{N}$. While block sizes may vary, each $s_i$ is assumed to be $\mathcal{O}(1)$. Let $\hat{\boldsymbol{x}}[B_i] \in \{0, 1\}^{s_i}$ denote the subvector for block $B_i$, i.e., the bits of $\hat{\boldsymbol{x}}$ that belong to that block.

**Templates and functions:** The transformation of the input vector depends on the configurations present within each block. Each block $B_i$ is associated with a finite set of templates $\mathcal{T}_i \subseteq \{0, 1\}^{s_i} \times \{0, 1\}^k$, where each template $(\boldsymbol{x}, \boldsymbol{y})$ maps a block configuration $\boldsymbol{x}$ to a complete output vector $\boldsymbol{y}$. The mapping is functional: for all $(\boldsymbol{x}, \boldsymbol{y}), (\boldsymbol{x}', \boldsymbol{y}') \in \mathcal{T}_i$, if $\boldsymbol{x} = \boldsymbol{x}'$, then $\boldsymbol{y} = \boldsymbol{y}'$. In other words, no block configuration maps to more than one output. Consequently, the cardinality of $\mathcal{T}_i$ is at most $2^{s_i}$

Using each $\mathcal{T}_i$, we define a block-specific pattern matching function $f_i : \{0, 1\}^{s_i} \to \{0, 1\}^k$ and an aggregation function $f : \{0, 1\}^k \to \{0, 1\}^k$ by:

$$f_i(\boldsymbol{x}') := \begin{cases} \boldsymbol{y} & \text{if } (\boldsymbol{x}', \boldsymbol{y}) \in \mathcal{T}_i \\ \boldsymbol{0} & \text{otherwise,} \end{cases} \qquad (5) \qquad\qquad f(\hat{\boldsymbol{x}}) := \bigvee_{i=1}^{b} f_i(\hat{\boldsymbol{x}}[B_i]) \qquad (6)$$

where $\boldsymbol{0}$ denotes the all-zero vector in $\{0, 1\}^k$, and the bitwise disjunction (logical OR) is applied elementwise across the output vectors $f_i(\hat{\boldsymbol{x}}[B_i])$ inside Equation (6). Each of these template matching functions is applied independently and simultaneously to the corresponding block.

In this framework, the block-specific templates $\mathcal{T}_i$ determine the local behavior of the algorithm, specifying how each block contributes to the global state vector $\hat{\boldsymbol{x}}$. The global update function $f$, formed by aggregating the output of all $f_i$, enforces this behavior across all blocks simultaneously. By applying $f$ iteratively, we propagate these local rules over time, effectively executing the algorithm.

## 4.1 Algorithmic example: computing binary addition

We now demonstrate how to apply the template matching principle to simulate binary addition. Throughout this example – and the more formal algorithm descriptions provided in Appendix B – we often assign descriptive variable names to improve clarity. These identifiers serve only as labels and do not affect computation. For this example, we denote the two summands by $p$ and $q$, each consisting of $\ell = 2$ bits. Consequently, their sum requires at most $\ell + 1 = 3$ bits to be represented.

The addition algorithm emulates a ripple-carry adder built from half-adders [10], performing bitwise addition, while propagating carries to higher-order bits. The process alternates between $\ell$ summation and $\ell$ carry-propagation steps, reaching a steady state after at most $2\ell$ iterations. However, as demonstrated in Appendix B.2, it is also possible to introduce a flag to indicate termination.

To simulate this behavior, the input structure is organized into blocks corresponding to the bits of $p$ and $q$, along with their associated carry bits. In this example, we use four blocks: two for the individual bits $p_i$ and $q_i$, and two others for the carries, denoted $c_i$, for $i = 1, 2$. In our representation, the final output comprises the most significant carry bit, followed by the bits of $p$ as stored in $\hat{\boldsymbol{x}}$. As shown in Figure 2, the result is formed by concatenating $c_2$, $p_2$, and $p_1$, for a total of $\ell + 1 = 3$ output bits. To capture the required operations in each block, we define a set of templates, as illustrated in Figure 2. The even-numbered templates implement bitwise summation, while the odd-numbered templates handle carry propagation to the next bit. These templates are designed to ensure that the operations proceed without interfering with one another.

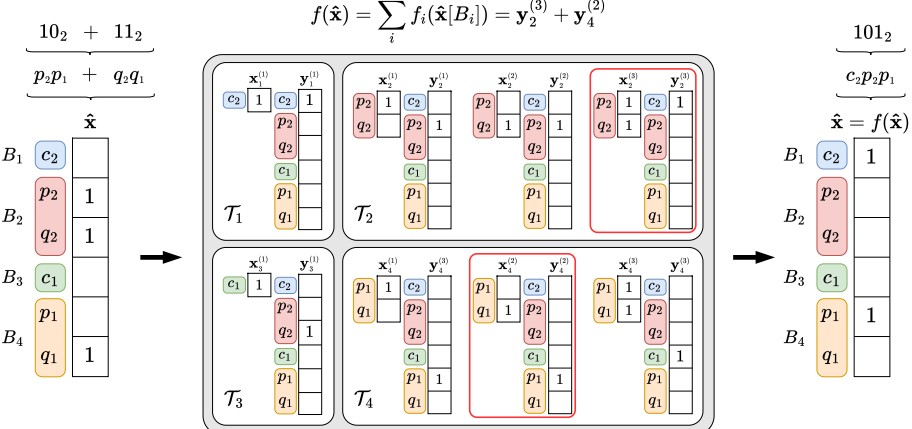

Figure 2: Illustration of the addition algorithm based on the template matching approach from Section 4. Two $\ell = 2$ bit numbers, $p = 2$ (or $10_2$) and $q = 3$ (or $11_2$), are added by organizing their bits and carries into blocks $B_i$. Blocks $B_2$ and $B_4$ represent the bits of $p$ and $q$, while $B_1$ and $B_3$ handle the carries. The input $\hat{\boldsymbol{x}}$ is processed via template matching $f$, using templates $\mathcal{T}_i$, producing outputs $y_i^{(k)}$ used to compose the output. Although the method is iterative, this example completes in one step. The final result 5 (or $101_2$) is stored at the most-significant carry bit and the bits of $p$ in $\hat{\boldsymbol{x}}$.

## 5 Exact learnability Part I: NTK predictor behavior

In this section, we analyze the exact learnability of algorithmic executions in neural networks by studying the NTK predictor, defined as the mean of the limiting distribution for a two-layer network. We show that it preserves sign-based information about the ground truth and can learn to execute algorithms framed as template matching, following the framework in Section 4. The training dataset is built from templates and, in the applications presented, its size scales with the number of bits, hence logarithmic in the number of possible binary inputs.

### 5.1 Input specification

We begin our analysis by specifying the structure of the training and testing inputs. Following the framework of Section 4, we construct the training inputs as block-partitioned vectors, where each block corresponds to an input template set. Each training input is non-zero only within the block determined by its associated input template in $\mathcal{T}_i$. In contrast, test inputs may contain multiple non-zero entries, each corresponding to a configuration appearing in the dataset. A visualization of the input configuration is provided in Figure 3.

**Training dataset** We define the training set for any algorithmic task described as in Section 4. Let each of the $b$ block configurations $\mathcal{T}_i$ be a set of $t_i := |\mathcal{T}_i|$ template-label tuples, i.e., each

$$\mathcal{T}_i = \{(\boldsymbol{x}^{(i,1)}, \boldsymbol{y}^{(i,1)}), \ldots, (\boldsymbol{x}^{(i,t_i)}, \boldsymbol{y}^{(i,t_i)})\} \subseteq \{0,1\}^{s_i} \times \{0,1\}^k.$$

The input dimension to the neural network is $k' = \sum_{i=1}^{b} t_i$.[3] We view $\mathbb{R}^{k'}$ as the direct sum $\mathbb{R}^{t_1} \oplus \cdots \oplus \mathbb{R}^{t_b}$ and for each subspace $\mathbb{R}^{t_i}$, we choose an orthonormal basis $\{\boldsymbol{u}_{i1}, \ldots, \boldsymbol{u}_{it_i}\}$. For each $i = 1, \ldots, b$, we encode the $j$-th template of $\mathcal{T}_i$, $(\boldsymbol{x}^{(i,j)}, \boldsymbol{y}^{(i,j)})$, as the block-partitioned vector

$$\boldsymbol{q}_{ij} = (\boldsymbol{0}_{t_1}, \ldots, \boldsymbol{0}_{t_{i-1}}, \boldsymbol{u}_{ij}, \ldots, \boldsymbol{0}_{t_b})^\top \in \mathbb{R}^{k'}$$

with the $i$-th block being equal to $\boldsymbol{u}_{ij}$. The sets of training inputs and corresponding ground-truth labels are given by

$$\mathcal{X} = \{\boldsymbol{q}_{ij} : i \in [b], j \in [t_i]\} \subseteq \mathbb{R}^{k'} \quad \text{and} \quad \mathcal{Y} = \{\boldsymbol{y}^{(i,j)} : i \in [b], j \in [t_i]\} \subseteq \mathbb{R}^k,$$

---

[3]In applications such as binary multiplication and SBN, the dimension $k'$ is extended with auxiliary unitary blocks, each containing a template. These extensions, detailed in Appendix C, ensure Assumption 5.1 is satisfied without affecting the algorithm, as they are never matched during execution.

$\mathcal{X}_{\text{init}}$     $\mathcal{X}$     $\mathcal{Y}$

Figure 3: Visualization of the input specification of Section 5.1 for binary summation of two $\ell = 2$ bit numbers. On the left, we illustrate the block structure of a pre-encoded test sample. Each block should either be zero or match the corresponding block of an element (row) of $\mathcal{X}_{\text{init}}$. On the left, we showcase the encoding procedure that creates the training dataset. The initial examples (described in Section 4) forming the rows of $\mathcal{X}_{\text{init}}$ are augmented and orthogonalized. Notice that the colored parts of each row of $\mathcal{X}_{\text{init}}$ along with the corresponding row of $\mathcal{Y}$ match the $\mathcal{T}_i$'s of Figure 2. Also note that the orthogonalization presented here is only one of the many possible ones. Finally, each row of $\mathcal{Y}$ depicts the corresponding ground-truth output for each training sample.

respectively. This encoding yields an orthonormal, block-partitioned set of training inputs, with no cross-block interference. In total, the dataset has size $k' \in \mathcal{O}(b)$, and $b$ depends on the number of bits used for number representation in each application. For example, to add two 10-bit numbers, 40 training examples are required.

**Test inputs** Based on the algorithmic execution framework of Section 4 every test input $\hat{\boldsymbol{x}}$ is expressed as $\hat{\boldsymbol{x}} = \frac{1}{\sqrt{n_{\hat{\boldsymbol{x}}}}}(\hat{\boldsymbol{x}}_1, \ldots, \hat{\boldsymbol{x}}_b)^\top$ with each $\hat{\boldsymbol{x}}_i$ being either $\boldsymbol{0}_{t_i}$ or matching one of the training samples of $\mathcal{X}$ in its $i$-th block, i.e. $\hat{\boldsymbol{x}}_i$ is equal to the $i$-th block of $\boldsymbol{q}_{ij}$ for some $j \in [t_i]$. In that case, we say that $\hat{\boldsymbol{x}}$ matches $\boldsymbol{q}_{ij}$. We denote by $n_{\hat{\boldsymbol{x}}}$ the number of blocks of $\hat{\boldsymbol{x}}$ that match an entry of the training set. Since each $t_i$ is $\mathcal{O}(1)$, the total number of test inputs is $\mathcal{O}(2^b)$, which is exponentially larger than the training dataset size. For instance, the total number of test inputs for the addition of two 10-bit numbers is $4^{10}$, which is exponentially larger than the training dataset size.

## 5.2 NTK predictor behavior theorem

To derive this result, we introduce and discuss one additional assumption. The specific structure of our orthogonal training set and test inputs simplifies the mean of the limiting distribution, $\mu(\hat{\boldsymbol{x}}) = \Theta(\hat{\boldsymbol{x}}, \mathcal{X})\Theta(\mathcal{X}, \mathcal{X})^{-1}\mathcal{Y}$. This predictor can be expressed as a simple weighted sum of the ground-truth training labels $\mathcal{Y}$. This sum is governed by two distinct weights: a "signal" weight, $w^1 \equiv w^1(\hat{\boldsymbol{x}})$, applied to training labels whose corresponding inputs match a block in the test input $\hat{\boldsymbol{x}}$, and an "interference" weight, $w^0 \equiv w^0(\hat{\boldsymbol{x}})$, applied to all labels from unmatched training inputs. For any given output bit, an "unwanted correlation" occurs when an unmatched training sample (which receives the $w^0$ weight) also has that bit set, thus contributing interference against the correct signal. Informally, we can interpret the ratio $-w^1/w^0$ as a decision margin. Our assumption, stated below, simply requires that the total number of these unwanted correlations is less than this margin, ensuring the signal's contribution outweighs the total interference. We now formally state the assumption that guarantees learnability:

**Assumption 5.1.** For each test input $\hat{\boldsymbol{x}}$ and for each position $i \in [k']$ such that the ground-truth output $f(\hat{\boldsymbol{x}})$[4] has the $i$-th bit set, the number of training examples that do not match $\hat{\boldsymbol{x}}$ and have the $i$-th bit set (which we call *unwanted correlations*) is less than the ratio $-w^1(\hat{\boldsymbol{x}})/w^0(\hat{\boldsymbol{x}})$.[5]

---

[4] There is a slight abuse of notation here when using $f(\hat{\boldsymbol{x}})$ since $f$ does not operate on encoded inputs. Depending on the context, we may use $\hat{\boldsymbol{x}}$ to denote both the pre-encoded and encoded test inputs.

[5] $w^0(\hat{\boldsymbol{x}})$ is always non-positive and so the ratio is non-negative.

While, at first, Assumption 5.1 may seem restrictive, we remark that for many algorithms, including those examined in this work, the number of conflicts is low enough to guarantee that it is satisfied. In particular, since the SBN instruction set is Turing-complete, it can, in principle, encode any algorithm. Our framework leverages this property by simulating SBN instructions, allowing any algorithm to be represented as input to the system. This does not mean that we directly learn arbitrary Turing-computable functions, but it ensures that such functions can be expressed within our formulation without violating Assumption 5.1. The behavior of the NTK predictor is given in the following theorem:

**Theorem 5.1** (NTK predictor behavior). *Consider an algorithmic problem cast as template-matching and encoded in a training set $(\mathcal{X}, \mathcal{Y}) \subseteq \mathbb{R}^{k'} \times \mathbb{R}^k$ as described in Section 5.1. Then, under Assumption 5.1, the mean of the limiting NTK distribution $\mu(\hat{\boldsymbol{x}}) = \Theta(\hat{\boldsymbol{x}}, \mathcal{X})\Theta(\mathcal{X}, \mathcal{X})^{-1}\mathcal{Y}$ for any test input $\hat{\boldsymbol{x}} \in \mathbb{R}^{k'}$ contains sign-based information about the ground-truth output, namely for each coordinate of the output $i = 1, \ldots, k$, $\mu(\hat{\boldsymbol{x}})_i \leq 0$ if the ground-truth bit at position $i$, $f(\hat{\boldsymbol{x}})_i$, is set, and $\mu(\hat{\boldsymbol{x}})_i > 0$ if the ground-truth bit at position $i$, $f(\hat{\boldsymbol{x}})_i$, is not set.*

*Proof outline.* We aim to express each $\mu(\hat{\boldsymbol{x}})_i$ as a weighted bit-sum with weights $w^0$ and $w^1$, and then rely on Assumption 5.1 to guarantee that the signs are preserved. The orthogonality of the training dataset forces the train NTK to align with the local computation structure of the template matching framework. In particular, both the train and test NTK kernels assume a scaled identity that keeps different blocks from interfering with one another. Furthermore, since any test input activates at most one vector per block, the test diagonal elements of the test NTK kernel (measuring the similarity of the test input to each element of the training set) take only two possible values: one indicating "this block is active" and one indicating "this block is empty". Concretely, $\Theta^{-1} := \Theta(\mathcal{X}, \mathcal{X})^{-1}$ takes the form $\tilde{\Theta}^{-1} \otimes I_k$ where $\tilde{\Theta}^{-1} \in \mathbb{R}^{k' \times k'}$ is a scaled identity plus a rank-1 perturbation (noise), and $\Theta(\hat{\boldsymbol{x}}, \mathcal{X})$ takes the form $\mathbf{f}^\top \otimes I_k$ where $\mathbf{f} \in \mathbb{R}^{k'}$ takes only two values, $\mathrm{f}^1$ and $\mathrm{f}^0$ denoting match/no-match. By arranging the elements of $\mathcal{Y}$ as columns in a matrix $Y$ and using vectorization, we can rewrite the model as

$$\mu(\hat{\boldsymbol{x}}) = \Theta(\hat{\boldsymbol{x}}, \mathcal{X})\Theta^{-1}\mathcal{Y} = ((\mathbf{f}\tilde{\Theta}^{-1}) \otimes I_k)\operatorname{vec}(Y) = Y\tilde{\Theta}^{-1}\mathbf{f}^\top.$$

We first show that $\tilde{\Theta}^{-1}\mathbf{f}^\top \in \mathbb{R}^{k'}$ takes only two values $w^0 \leq 0$ and $w^1 > 0$ depending on whether the corresponding entry in $\mathbf{f}$ is equal to $\mathrm{f}^0$ or $\mathrm{f}^1$. To conclude, we write

$$\mu(\hat{\boldsymbol{x}})_i = \sum_{(j,l)\in\mathcal{I}_+(\hat{\boldsymbol{x}})} f(\boldsymbol{q}_{jl})_i w^1 + \sum_{(j,l)\in\mathcal{I}_-(\hat{\boldsymbol{x}})} f(\boldsymbol{q}_{jl})_i w^0, \tag{7}$$

where

$$\mathcal{I}_+(\hat{\boldsymbol{x}}) = \{(j,l) : j \in [m], l \in [s_j] \text{ and } \hat{\boldsymbol{x}} \text{ matches } \boldsymbol{q}_{jl}\},$$

and

$$\mathcal{I}_-(\hat{\boldsymbol{x}}) = \{(j,l) : j \in [m], l \in [s_j] \text{ and } \hat{\boldsymbol{x}} \text{ doesn't match } \boldsymbol{q}_{jl}\}.$$

The sets $\mathcal{I}_+(\hat{\boldsymbol{x}})$ and $\mathcal{I}_-(\hat{\boldsymbol{x}})$ partition the training dataset into two disjoint sets: the indices of the training dataset that match $\hat{\boldsymbol{x}}$ and the ones that do not. When $f(\hat{\boldsymbol{x}})_i = 0$, Equation (6) yields a vanishing first summation and therefore $\mu(\hat{\boldsymbol{x}})_i \leq 0$. On the other hand, when $f(\hat{\boldsymbol{x}})_i = 1$, due to the fact that $\hat{\boldsymbol{x}}$ matches exactly one of the training samples from the block containing the $i$-th bit, Equation (7) reduces to $\mu(\hat{\boldsymbol{x}})_i = w^1 + |\mathcal{I}_-^1(\hat{\boldsymbol{x}})| \cdot w^0$, where $\mathcal{I}_-^1(\hat{\boldsymbol{x}}) = \{(j,l) \in \mathcal{I}_-(\hat{\boldsymbol{x}}) : f(\boldsymbol{q}_{jl})_i = 1\}$ denotes the index set of unwanted correlations. Under Assumption 5.1, we have $\mu(\hat{\boldsymbol{x}})_i > 0$ and so the sign-based ground truth information is preserved, concluding the proof. A pictorial version of the above outline is given in Figure 4.

$\square$

Regarding the algorithms discussed throughout this paper, we have the following remark[6]:

**Remark 5.1.** *The tasks of* binary permutations*,* binary addition*,* binary multiplication*, and* executing SBN instructions *all satisfy the assumptions of Theorem 5.1.*

---

[6]Our results for permutation, addition, and multiplication were numerically verified on various random instances by calculating the corresponding limiting mean. For code and implementation details, refer to the supplementary material.

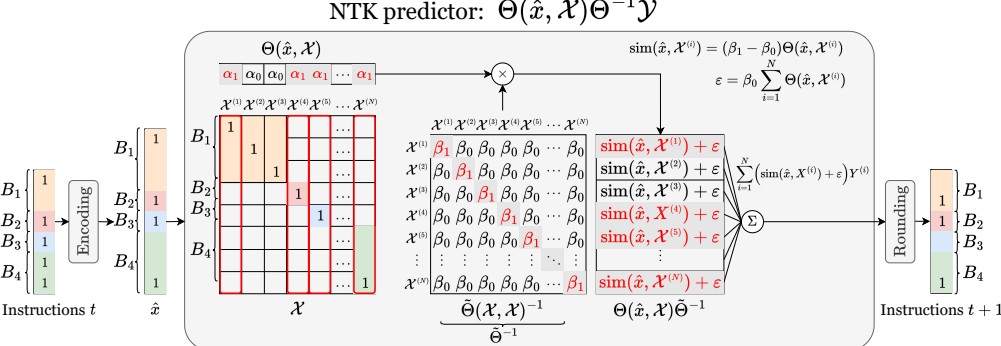

NTK predictor: $\Theta(\hat{x}, \mathcal{X})\tilde{\Theta}^{-1}\mathcal{Y}$

Figure 4: Illustration of the NTK predictor structure: inputs are first encoded (normalization omitted) to compute the test NTK $\Theta(\hat{x}, \mathcal{X})$. Due to the test input structure, this kernel assumes two values based on matches between test and training inputs within blocks. Multiplying by $\tilde{\Theta}^{-1}$ (which assumes the form of scaled identity plus a rank-1 noise perturbation colored black) re-weights these similarities, and the multiplication by $\mathcal{Y}$ gives the final prediction. When the contribution of the unmatched entries is controlled (black similarities), the sign of each coordinate matches the ground-truth output.

The proof strategy for Remark 5.1 involves computing a lower bound for the decision margin $-w^1(\hat{\boldsymbol{x}})/w^0(\hat{\boldsymbol{x}})$ over all test inputs $\hat{\boldsymbol{x}}$ and showing that the number of unwanted correlations falls below this minimum value. For example, in the case of binary addition, this threshold comes out to be equal to $4$ while the number of unwanted correlations can be at most $1$. In Figure 3, this is captured by the fact that each column of $\mathcal{Y}$ contains at most $2$ ones. A complete proof of Theorem 5.1 and Remark 5.1 can be found in Appendix C.

## 6 Exact learnability Part II: high-probability guarantee

In this section, we continue our proof of exact learning of algorithmic instructions using neural networks. The conclusion of Theorem 5.1 suggests a simple procedure: for each coordinate, if the output of the network is greater than zero, round to $1$, otherwise round to $0$. To extend this result from the NTK predictor to actual models and ensure high-probability guarantees of exact learning, we can independently train enough models, average their outputs, and round accordingly.[7] We define *ensemble complexity* as the number of models required to achieve a desired level of post-rounding accuracy. In what follows, we derive a lower bound on the ensemble complexity of learning algorithmic instructions and give its asymptotic order. This completes the proof that neural networks can, with high probability, exactly learn algorithmic instructions.

Given a test input $\hat{\boldsymbol{x}}$ with ground truth $\hat{\boldsymbol{y}} = f(\hat{\boldsymbol{x}})$, let $F^j(\hat{\boldsymbol{x}})$ be the output of the $j$-th model in an ensemble of $N$ independently trained networks. By Theorem 3.1, each $F^j(\hat{\boldsymbol{x}})$ is drawn i.i.d. from $\mathcal{N}(\mu(\hat{\boldsymbol{x}}), \Sigma(\hat{\boldsymbol{x}}))$, so every coordinate $F^j(\hat{\boldsymbol{x}})_i$ follows $\mathcal{N}(\mu(\hat{\boldsymbol{x}})_i, \sigma^2(\hat{\boldsymbol{x}}))$.[8] Define the ensemble mean $G(\hat{\boldsymbol{x}}) = \frac{1}{N}\sum_{j=1}^{N} F^j(\hat{\boldsymbol{x}})$. Because $\mu(\hat{\boldsymbol{x}})_i \leq 0$ when $\hat{\boldsymbol{y}}_i = 0$ and $\mu(\hat{\boldsymbol{x}})_i > 0$ when $\hat{\boldsymbol{y}}_i = 1$, rounding $G(\hat{\boldsymbol{x}})_i$ is correct if $|G(\hat{\boldsymbol{x}})_i - \mu(\hat{\boldsymbol{x}})_i| < |\mu(\hat{\boldsymbol{x}})_i|/2$.

Applying a standard Gaussian concentration bound given in Lemma A.1 and the union bound, we obtain, for any $\delta \in (0, 1)$, perfect post-rounding accuracy with probability $1-\delta$ whenever the number of averaged models $N$ satisfies:

$$N \geq 8 \max_{\hat{\boldsymbol{x}}, i \in [k]} \left\{ \frac{\sigma^2(\hat{\boldsymbol{x}})}{\mu^2(\hat{\boldsymbol{x}})_i} \right\} \ln\left(\frac{2k'}{\delta}\right). \tag{8}$$

---

[7]In practice, model-to-model variability is often controlled by training several copies (or by collecting multiple checkpoints along one training run) and then averaging either their predictions [17] or their weights [13]. Our "ensemble complexity" result gives a clean theoretical analogue of this variance-reduction trick, with concrete high-probability guarantees on post-rounding accuracy.

[8]See Appendix D for the calculation of $\sigma^2(\hat{\boldsymbol{x}})$.

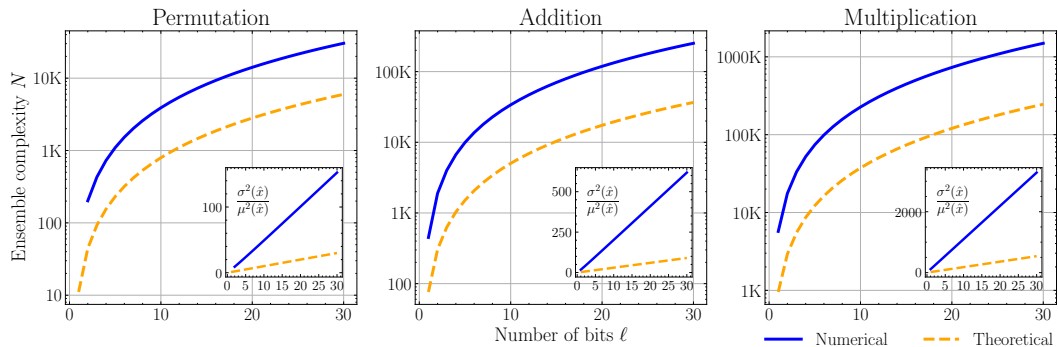

Figure 5: Numerical and theoretical estimates of ensemble complexity $N$ (in log-scale) for permutation, addition, and multiplication tasks as a function of bit length $\ell$. Ensemble complexity is computed via a union bound over all possible inputs and algorithmic executions for a given $\ell$. Inset blocks illustrate the ratio of variance to mean in Equation (8), estimated using input size $k'$. This ratio increases linearly with $k'$ up to a constant. The same ratio is used in the theoretical estimate of $N$, which matches the numerical estimate in growth rate, differing only by a constant factor.

To derive the order of the bound on the ensemble complexity of Equation (8), we need to analyze the asymptotic orders of $\mu(\hat{\boldsymbol{x}})$ and $\sigma^2(\hat{\boldsymbol{x}})$. The result is summarized in the following technical lemma:

**Lemma 6.1.** *Suppose we train on any of the four tasks (permutations, addition, multiplication, SBN instructions), as described in Section 5. Let $\hat{\boldsymbol{x}}$ be a test input unseen during training matching $n_{\hat{\boldsymbol{x}}} > 1$ training entries and let $\mu(\hat{\boldsymbol{x}})$ and $\sigma^2(\hat{\boldsymbol{x}})$ be as in Theorem 3.1. Then $\sigma^2(\hat{\boldsymbol{x}}) \in \mathcal{O}(1/k')$, and, for each $i \in [k]$, depending on the relationship between $n_{\hat{\boldsymbol{x}}}$ and $k'$ we have:*

$$|\mu(\hat{\boldsymbol{x}})_i| \in \begin{cases} \Theta\left(\frac{1}{k'}\right) & \text{if } n_{\hat{\boldsymbol{x}}} \text{ is const.} \\ \Theta\left(\frac{\sqrt{n_{\hat{\boldsymbol{x}}}}}{k'}\right) & \text{if } n_{\hat{\boldsymbol{x}}} \text{ is non-const.} \\ & \text{and sublinear in } k' \\ \Theta\left(\frac{1}{\sqrt{k'}}\right) & \text{if } n_{\hat{\boldsymbol{x}}} = ck' \text{ for} \\ & \text{some } c \in (0,1] \end{cases} \quad \text{and} \quad |\mu(\hat{\boldsymbol{x}})_i| \in \begin{cases} \Theta(1) & \text{if } n_{\hat{\boldsymbol{x}}} \text{ is const.} \\ \Theta\left(\frac{1}{\sqrt{n_{\hat{\boldsymbol{x}}}}}\right) & \text{if } n_{\hat{\boldsymbol{x}}} \text{ is non-const.} \\ & \text{and sublinear in } k' \\ \Theta\left(\frac{1}{\sqrt{k'}}\right) & \text{if } n_{\hat{\boldsymbol{x}}} = ck' \text{ for} \\ & \text{some } c \in (0,1) \\ \Theta\left(\frac{1}{\sqrt{k'}}\right) & \text{if } n_{\hat{\boldsymbol{x}}} = k \text{ and there} \\ & \text{are unwanted corr.} \\ \Theta\left(\frac{1}{k'}\right) & \text{if } n_{\hat{\boldsymbol{x}}} = k' \text{ and there} \\ & \text{are no unwanted corr.} \end{cases}$$

*when the ground-truth bit at position $i$ is not set ($f(\hat{\boldsymbol{x}})_i = 0$) or set ($f(\hat{\boldsymbol{x}})_i = 1$), respectively.*

The proof of Lemma 6.1 (including the calculation of $\sigma^2(\hat{\boldsymbol{x}})$) can be found in Appendix D. In light of these asymptotic results, the uniform bound of Equation (8) behaves like $\mathcal{O}(k' \log k')$. An application of the union bound shows that for an algorithm requiring $m$ iterations, the ensemble complexity bound when accounting for all $\mathcal{O}(2^b)$ possible test inputs (where $b$ is the number of templates as in Section 4) and all $m$ iterations behaves like $\mathcal{O}(k'b + k' \log m)$. In particular, for the tasks considered, we have the following remark:

**Remark 6.1.** *For the tasks of binary permutation ($b = k' = \ell$, $m = 1$), binary addition ($b = 2\ell$, $k' = 4\ell$, $m = 2\ell$), and binary multiplication ($b = 11\ell$, $k' = 21\ell$, $m = 4\ell^2 + 3\ell$) the ensemble complexity scales like $\mathcal{O}(\ell^2)$, where $\ell$ is the bit length for each application.*

Figure 5 plots numerical and theoretical estimates for the ensemble complexity of the permutation, addition, and multiplication tasks, showcasing the conclusion of Remark 6.1. In Appendix E, we verify our theory with an empirical estimate of the ensemble complexity for the permutation task. This is done by training multiple two-layer fully connected feed-forward networks, each with 50,000 hidden units, using full-batch gradient descent to form an ensemble.

# 7  Limitations and future work

In this work, we have demonstrated that two-layer fully connected feed-forward neural networks in the infinite-width limit can learn to execute algorithmic instructions expressed within our template matching framework (including binary permutations, binary addition, binary multiplication, and execution of SBN instructions) using a training set of logarithmic size in the number of possible binary inputs. This provides an affirmative answer to the question of whether neural networks can learn to execute long sequences of binary-encoded instructions exactly.

Our analysis, however, relies on several simplifying assumptions that bound its generality. The first concerns data orthogonality and explicit instruction access, which guarantee that each local computation step is independently learnable. Future work could investigate whether exact learning remains possible under correlated training examples or when the network must infer primitive instructions from only partial input–output traces. The second limitation arises from the bounded memory setting of our framework. Increasing the available memory changes the input dimensionality of the model and, therefore, requires retraining, so extrapolation to longer inputs does not occur automatically. Within this bounded memory regime, exact algorithmic execution remains achievable using only logarithmically many short training examples, but extending these results to architectures that naturally process variable-length inputs, such as RNNs, Transformers, or GNNs, would be a valuable next step. For example, the use of GNNs on bounded degree graphs may enable controlled forms of length generalization with respect to graph size while preserving the theoretical structure of our exact learning framework.

## Acknowledgments

K. Fountoulakis would like to acknowledge the support of the Natural Sciences and Engineering Research Council of Canada (NSERC). Cette recherche a été financée par le Conseil de recherches en sciences naturelles et en génie du Canada (CRSNG), [RGPIN-2019-04067, DGECR-2019-00147].

G. Giapitzakis would like to acknowledge the support of the Onassis Foundation - Scholarship ID: F ZU 020-1/2024-2025.

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

# A Notation and Preliminaries

For two matrices $A_1 \in \mathbb{R}^{n_1 \times m_1}$, $A_2 \in \mathbb{R}^{n_2 \times m_2}$ we denote by $A_1 \otimes A_2 \in \mathbb{R}^{n_1 n_2 \times m_1 m_2}$ their Kronecker product. It is relatively easy to show that when $A$ and $B$ are square matrices (i.e. $n_1 = m_1$ and $n_2 = m_2$), the eigenvalues of $A_1 \otimes A_2$ are given exactly by the products of the eigenvalues of $A_1$ and $A_2$. In particular, $A_1 \otimes A_2$ is positive definite if $A_1$ and $A_2$ are positive definite.

## A.1 Useful Results

In this section, we state two results from linear algebra and probability theory that are used to derive our main results. The first result is used to compute the train NTK matrix:

**Theorem A.1** (Sherman and Morrison 33). *Suppose $A \in \mathbb{R}^{n \times n}$ is an invertible matrix and $\boldsymbol{u}, \boldsymbol{v} \in \mathbb{R}^n$. Then $A + \boldsymbol{u}\boldsymbol{v}^\top$ is invertible if and only if $1 + \boldsymbol{v}^\top A^{-1} \boldsymbol{u} \neq 0$. In this case,*

$$(A + \boldsymbol{u}\boldsymbol{v}^\top)^{-1} = A^{-1} - \frac{A^{-1} \boldsymbol{u}\boldsymbol{v}^\top A^{-1}}{1 + \boldsymbol{v}^\top A^{-1} \boldsymbol{u}}$$

The second result is a concentration bound for sums of Gaussian random variables, which we used to derive our ensemble complexity bounds:

**Lemma A.1.** *Let $X_1, X_2, \ldots, X_n$ be independent Gaussian random variables with mean $\mu$ and variance $\sigma^2$ and let $\overline{X}_n = \sum_{i=1}^n X_i$. Then for all $t > 0$, we have:*

$$\mathbb{P}\left\{|\overline{X}_n - \mu| \geq t\right\} \leq 2\exp\left(-\frac{nt^2}{2\sigma^2}\right)$$

*Proof.* We will use the Chernoff technique. Let $\lambda > 0$. We have $\overline{X}_n - \mu \sim \mathcal{N}\left(0, \frac{\sigma^2}{n}\right)$ and so by symmetry and Markov's inequality we get:

$$\mathbb{P}\left\{|\overline{X}_n - \mu| \geq t\right\} = 2\mathbb{P}\left\{\overline{X}_n - \mu \geq t\right\} = 2\mathbb{P}\left\{e^{\lambda(\overline{X}_n - \mu)} \geq e^{\lambda t}\right\} \leq e^{-\lambda t} \cdot \mathbb{E}\left[e^{\lambda(\overline{X}_n - \mu)}\right] \quad (9)$$

The expectation on the right-hand side is equal to the moment-generating function of a normal distribution with mean 0 and variance $\sigma^2/n$ and so it is equal to $\exp\left(\frac{\sigma^2 \lambda^2}{2n}\right)$. Now let

$$\phi(\lambda) = \exp\left(\frac{\sigma^2 \lambda^2}{2n} - \lambda t\right)$$

be the right-hand side of Equation (9). Minimizing $\phi(\lambda)$ with respect to $\lambda$ we find that the minimum occurs at $\lambda^* = \frac{nt}{\sigma^2}$ and plugging this back into Equation (9) gives the required bound. $\square$

# B Constructive proofs

In this section, we outline the set of instructions used to illustrate the steps involved in the algorithms described earlier. To ensure clarity and avoid unnecessary repetition, we adopt certain conventions in the presentation of these instructions.

To ensure the correctness of our constructive proofs for the numerical tasks presented below, we include a numerical validation.[9] This validation uses the instructions defined below for permutation, addition, and multiplication. Using the Neural Tangents package [25], we compute the NTK predictor (as in Theorem 3.1). Applying the encoding and rounding procedures described in Section 5 and Section 6, we demonstrate that the implementations are numerically correct for bit lengths up to $\ell = 10$. Additionally, we provide a demonstration script that more descriptively illustrates each step of the algorithms as executed within our framework.

---

[9]Our source code can be found in the supplementary material.

**Conventions**

Unless otherwise stated, all variables refer to Boolean values (i.e., elements of $\{0,1\}$), arrays of Boolean values, or natural numbers as appropriate. Let $\mathrm{Index}(A)$ denote the index set of an array $A$.

- **Logical equivalence:** For Boolean variables $A$ and $B$,

$$A = B \iff^{\text{def}} (A = 1 \text{ AND } B = 1) \text{ OR } (A = 0 \text{ AND } B = 0)$$

  This represents equality of Boolean values, not assignment.

- **Logical inequality:**

$$A \neq B \iff^{\text{def}} (A = 1 \text{ AND } B = 0) \text{ OR } (A = 0 \text{ AND } B = 1)$$

- **Assignment:** We use the symbol $\leftarrow$ to denote assignment. For Boolean variables:

$$B \leftarrow A \quad \text{means that } B \text{ is assigned the current value of } A$$

- **Universal indexing:**

$$A[\text{ALL}] \iff^{\text{def}} \forall i \in \mathrm{Index}(A), \ A[i]$$

- **Bitwise comparison:**

$$A[\text{ALL}] = B[\text{ALL}] \iff^{\text{def}} \forall i \in \mathrm{Index}(A) : A[i] = B[i]$$

- **Bitwise assignment:**

$$A[\text{ALL}] \leftarrow v \iff^{\text{def}} \forall i \in \mathrm{Index}(A) : A[i] \leftarrow v$$

- **Binary representation:** Let $\mathrm{bin}(v)$ denote the binary representation of a natural number $v$, encoded as a Boolean array. The bit width is inferred from the context unless specified.

## B.1  Binary permutation

In this section, we demonstrate how the template matching framework can be applied to execute binary permutations. Let the binary input be denoted by a binary number p. We begin by defining the input structure in terms of blocks and then construct the corresponding templates.

**Blocks**: For an $\ell$-bit number p, we design $\ell$ blocks, each encoding a single bit of the number. Let each block correspond to a bit p[i], thus we have:

- (p[i]) for $i \in [\ell]$

**Instructions**: Given this block structure, we now define the instructions that encode the desired permutation. Consider a mapping $\pi : [\ell] \to [\ell]$ that specifies the permutation: it takes a bit position as input and returns its new position after the permutation. For example, if the third bit of the input is to be moved to the fifth bit position, then $\pi(3) = 5$.

Based on this mapping, we construct $\ell$ samples, one for each block, which encodes the transformation defined by $\pi$.

**Instructions: Permutation**

```
INPUT: x[p[i]] = 1
OUTPUT: y[p[π(i)]] ← 1
Instruction count: ℓ
```

Each bit permutation is thus encoded as an individual instruction in the template set. This captures the behavior that when a specific bit position is activated in the input, its permuted position must also be activated in the output.

## B.2 Binary addition

With the established framework, we now illustrate how to apply the template matching principle to simulate binary addition. Throughout this and other algorithmic examples, we often assign descriptive variable names to improve clarity. These identifiers serve only as labels and do not affect computation.

Let the binary inputs be denoted p and q, each consisting of $\ell$ bits. The result of their sum requires $\ell + 1$ bits. We begin by defining the block structure of $x$.

**Blocks:** In this implementation, we organize the input into $2\ell$ blocks, $\ell$ blocks encode the bits of p and q, and the $\ell$ blocks encode the corresponding carry bits c.

- (p[i], q[i]) for $i \in [\ell]$
- (c[i]) for $i \in [\ell]$

Assignments and conditions are written using square bracket notation. Since each block comprises uniquely named variables, individual variables can be referenced directly by name. For example, setting the second carry variable to 1 is expressed as x[c[2]] $\leftarrow$ 1.

The addition algorithm follows a ripple-carry approach using half-adders [10]. It proceeds in two alternating phases: bitwise summation and carry propagation.

In the summation phase, the algorithm adds the bits x[p[i]] and x[q[i]] for each $i$, storing the result back in x[p[i]] and placing any resulting carry in x[c[i]]. In the subsequent carry propagation phase, the carry x[c[i]] is transferred to x[q[i+1]], allowing it to participate in the next summation step.

This iterative process alternates between $\ell$ summation steps and $\ell$ carry propagation steps. After $2\ell$ iterations, the computation reaches a steady state.

In our representation, the final output consists of the carry of the most-significant bit y[c[$\ell$]] concatenated with all bits $i \in \{\ell, \ldots, 1\}$ in y[p[i]], yielding a total of $\ell + 1$ output bits.

The following instructions are purposely designed to minimize the number of unwanted correlations when using the NTK predictor. Specifically, the highest number of such correlations (also referred to as conflicts) occurs in the coordinates encoding the bits of the summand p[i] and the most significant carry bit c[$\ell$], where two instructions share a non-zero entry for the same coordinate. Notably, however, the maximum number of conflicts per coordinate – equal to 1 in this case – remains constant and does not increase with the bit count $\ell$. This bounded conflict rate enables learnability, as further discussed in Appendix C.

**Instructions:** To capture the aforementioned processes in the blocks, we define a set of representative instructions. By convention, we assume that any variable not explicitly set in the output is assigned a value of zero.

**Instructions: Bitwise addition**

```
INPUT: x[p[i]] = 0 AND x[q[i]] = 1
OUTPUT: y[p[i]] ← 1
Instruction count: ℓ
```

```
INPUT: x[p[i]] = 1 AND x[q[i]] = 0
OUTPUT: y[p[i]] ← 1
Instruction count: ℓ
```

```
INPUT: x[p[i]] = 1 AND x[q[i]] = 1
OUTPUT: y[c[i+1]] ← 1
Instruction count: ℓ
```

The case where both x[p[i]] and x[q[i]] are zero does not require an instruction. Since no template matches, the default behavior results in all outputs being zero for that block and its carry, which is consistent with expected addition logic.

**Instructions: Carry propagation**

```
INPUT: x[c[i]] = 1
OUTPUT: (i < ℓ): y[q[i+1]] ← 1
        (i = ℓ): y[c[i]] ← 1
Instruction count: ℓ
```

This defines the *carry-propagation* behavior: when the carry at position $i$ is one, its effect is passed to the next summand block.

An important observation is that template matching across different blocks does not interfere between phases. During bitwise summation, the carry blocks are set to zero and thus remain inactive. Conversely, during carry propagation, all x[q[i]] entries are empty, so summation remains static, allowing the carry from x[c[i-1]] to be transmitted to x[q[i]] without conflict.

Finally, note that once x[c[ℓ]] becomes non-zero, it remains set for the remainder of the algorithm. This value represents the most significant bit (MSB) of the final result.

**Termination:** As previously mentioned, this implementation reaches a steady state after $2\ell$ iterations. However, it is also possible to introduce a termination flag that is triggered once a specific condition is met. In this context, we define a termination flag that becomes active once $2\ell$ iterations have elapsed. To implement this, we introduce the following additional blocks:

- (counter[i]) for $i \in [2\ell]$

**Instructions: termination**

```
INPUT: x[counter[i]] = 1
OUTPUT: (if i < 2ℓ): y[counter[i+1]] ← 1
        (if i = 2ℓ): y[counter[i]] ← 1
```

This design introduces $2\ell$ additional blocks. Once the first block is activated, each block subsequently activates the next, until x[counter[2ℓ]] is reached. The activation of x[counter[2ℓ]] indicates that the algorithm has finished.

## B.3 Binary multiplication

In this section, we describe the structure and the components used to perform binary multiplication between two $\ell$-bit binary variables. Our implementation simulates the shift-and-add multiplication algorithm [10].

In essence, the shift-and-add algorithm multiplies two binary numbers by scanning each bit of the multiplier. If a bit is $1$, the appropriately shifted multiplicand is added to the running total. This method mirrors the principle of long multiplication, but relies solely on shifts and additions rather than full multiplications.

To implement this algorithm, we divide it into four distinct processes:

1. Check the least significant bit (LSB) of the multiplier
2. Add multiplicand to the accumulating total
3. Copy the multiplicand to the addition scratchpad
4. Shift multiplicand and multiplier

Each of these processes is implemented using one or more blocks in the input. Because of how the algorithm operates, we represent the multiplicand using $2\ell$ bits, initializing the most significant $\ell$ bits to zero. The bits are stored in little-endian form, so the first entry represents the least significant bit (LSB).

**Blocks:** We begin by defining the block structure:

- (multiplier[1], to_shift_right[1], to_check_lsb)

- `(multiplier[i], to_shift_right[i])` for $i \in [2, \ell]$
- `(multiplicand[i], to_shift_left[i], to_copy_to_sum_q[i])` for $i \in [2\ell]$
- `(sum_p[i], sum_q[i])` for $i \in [2\ell]$
- `(sum_c[i])` for $i \in [2\ell]$
- `(sum_counter[i])` for $i \in [4\ell]$

The first group of blocks stores the multiplier bits along with their associated shift flags. Flags do not hold data themselves. Instead, they signal when a specific action should be triggered. These are typically (though not exclusively) prefixed with `to_`. The block of the least significant bit of the multiplier includes an additional flag for checking its value.

The multiplicand bits are also paired with their shift flags and an extra flag for copying. This copying flag signals when to transfer the multiplicand into the summation scratchpad.

The remaining blocks represent the summation components, as introduced in Appendix B.2.

Therefore, in total, there are $11\ell$ blocks. However, during execution, at most $7\ell + 1$ can be active at any time during execution. This restriction arises from the counter blocks, where only one block can be active at a time.

The result of the multiplication is a $2\ell$-bit number, stored in `x[sum_p]`[ALL]. Each iteration of the algorithm may involve one or more of the four described processes, and some processes themselves may span multiple iterations.

- Addition: $4\ell$ iterations
- Check least significant bit (LSB): 1 iteration
- Copy multiplicand: 1 iteration
- Shift multiplicand to the left and multiplier to the right (simultaneously): 1 iteration

The worst-case runtime occurs when the multiplier consists entirely of 1s, triggering all operations in each cycle. In this case, the algorithm performs $\ell$ full iterations, resulting in a total execution count of $4\ell^2 + 3\ell$.

The following instructions exhibit a finite number of conflicts per coordinate, a crucial property for ensuring NTK learnability, as discussed in more detail in Appendix C. These conflicts can be quantified by counting the number of outputs that share the same coordinate. In the case of multiplication, the number of conflicts per coordinate is 2. This occurs in the coordinates corresponding to the `multiplier` and `multiplicand` bits, which may be either modified or preserved depending on the values of the flags within their respective blocks.

**Instructions**  Based on processes and the block structure previously defined, we now define the binary instructions for each of the processes and their corresponding blocks to perform binary multiplication.

**Instructions: preserve multiplier and multiplicand**

While any other secondary process is being executed, the values in the multiplier and multiplicand bits must be preserved. For that, we define:

```
INPUT: x[multiplier[1]] = 1 AND x[to_shift_right[1]] = 0 AND
       x[to_check_lsb] = 0
OUTPUT: y[multiplier[1]] ← 1
Instruction count: 1
```

```
INPUT: x[multiplier[i]] = 1 AND x[to_shift_right[i]] = 0
OUTPUT: (i > 1) y[multiplier[i]] ← 1
Instruction count: ℓ − 1
```

```
INPUT: x[multiplicand[i]] = 1 AND x[to_shift_left[i]] = 0
OUTPUT: y[multiplicand[i]] ← 1
Instruction count: 2ℓ
```

**Instructions: check least significant bit**

For this stage, we have to define two instructions for when the `to_check_lsb` flag is activated. If the LSB of the multiplier is equal to one, then we start the addition process by activating the flags to copy the multiplicand to the addition stage. In contrast, if the LSB is zero, we trigger the shifting process of both multiplicand and multiplier.

```
INPUT: x[multiplier[1]] = 1 AND x[to_shift_right[1]] = 0 AND
       x[to_check_lsb] = 1
OUTPUT: y[multiplier[1]] ← 1 AND y[to_copy_to_sum_q[ALL]] ← 1
Instruction count: 1
```

and

```
INPUT: x[multiplier[1]] = 0 AND x[to_shift_right[1]] = 0 AND
       x[to_check_lsb] = 1
OUTPUT: y[to_shift_right[ALL]]←1 AND y[to_shift_left[ALL]]←1
Instruction count: 1
```

**Instructions: copy multiplicand to addition block**

Once the multiplicand copy flag is activated, we have to send the data to the `sum_q[i]` variable. While the copying should only cover the cases for which the multiplicand bit is equal to one, we have an extra functionality for the first bit of the multiplicand, which triggers the counter to start the addition process. Because of this, we require an additional instruction that also covers the case when `x[multiplicand[1]] = 0`.

```
INPUT: x[multiplicand[1]] = 0 AND x[to_copy_to_sum_q[1]] = 1
OUTPUT: y[sum_counter[1]] ← 1
Instruction count: 1
```

```
INPUT: x[multiplicand[i]] = 1 AND x[to_copy_to_sum_q[i]] = 1
OUTPUT: (i = 1) y[multiplicand[i]] = 1 AND y[sum_q[i]] ← 1 AND
                y[sum_counter[i]] ← 1
        (i > 1) y[multiplicand[i]] = 1 AND y[sum_q[i]] ← 1
Instruction count: ℓ
```

**Instructions: add multiplicand to the running total**

For this operation, we define the same instructions that were defined in Appendix B.2 for the variables p and q, which have $2\ell$ bits in this context. These instructions cover the all the processes of bitwise addition, carry propagation, and counter update. By the end of the addition process, signalled by `x[sum_counter[2ℓ]] = 1`, we activate the shift process in the multiplicand and multiplier.

```
INPUT: x[sum_p[i]] = 1 AND x[sum_q[i]] = 0
OUTPUT: y[sum_p[i]] ← 1
Instruction count: 2ℓ
```

```
INPUT: x[sum_p[i]] = 0 AND x[sum_q[i]] = 1
OUTPUT: y[sum_p[i]] ← 1
Instruction count: 2ℓ
```

```
INPUT: x[sum_p[i]] = 1 AND x[sum_q[i]] = 1
OUTPUT: y[sum_c[i]] ← 1
Instruction count: 2ℓ
```

```
INPUT: x[sum_c[i]] = 1
OUTPUT: (i < 2ℓ) y[sum_q[i+1]] ← 1
Instruction count: 2ℓ − 1
```

```
INPUT: x[sum_counter[i]] = 1
OUTPUT: (i < 4ℓ) y[sum_counter[i+1]] ← 1
        (i = 4ℓ) y[to_shift_right[ALL]] ← 1 AND y[to_shift_left[ALL]] ← 1
Instruction count: 4ℓ
```

### Instructions: shift multiplier to the right

The purpose of this function is to perform the following behavior: when the `to_shift_right` flag is active, bit $i$ of `multiplier` is assigned the value of the previous bit. If `multiplier` is already 0, it is set to zero directly without further computation. An exception is made for $i = 1$: it does not shift its value but triggers the `to_check_lsb` flag whenever `to_shift_right[1]` is active, regardless of the corresponding `multiplier` bit.

```
INPUT: x[multiplier[i]] = 0 AND x[to_shift_right[i]] = 1
OUTPUT: (i = 1) y[to_check_lsb] ← 1
Instruction count: 1
```

```
INPUT: x[multiplier[i]] = 1 AND x[to_shift_right[i]] = 1
OUTPUT: (i = 1) y[to_check_lsb] ← 1
        (i > 1) y[multiplier[i-1]] ← 1
Instruction count: ℓ
```

### Instructions: shift multiplicand to the left

The goal of these instructions is to execute the following Instructions: when the `to_shift_left` flag is active, shift the `multiplicand` by assigning each bit to the next lower-order position. If $i = 2ℓ$, or if `multiplicand` is already 0, set the value to zero directly, as the shift is implicitly handled.

```
INPUT: x[multiplicand[i]] = 1 AND x[to_shift_left[i]] = 1
OUTPUT: (i < 2ℓ) y[multiplicand[i+1]] ← 1
Instruction count: 2ℓ − 1
```

### B.4   General computation

In this subsection, we present results that address the generality of the template matching approach previously described in Section 4. To this end, we demonstrate that we can build a block structure and corresponding instructions to simulate a one-instruction set computer (OISC), thereby showing that we can execute any computable function, provided with the right instructions and memory values. More specifically, in this proof, we represent an OISC with a single instruction called "Subtract and branch if negative" or SBN [8].

**SBN:** Named for its operation "subtract and branch if negative", SBN is a one-instruction set computer. One way to express SBN is detailed in Algorithm 1, and it consists of subtracting the content at address $a$ from that at address $b$, and storing the result back at $b$. All these values are stored in a memory array. If the result is positive, the computer executes the next instruction; otherwise, it jumps to the instruction at address $c$. Despite this operational simplicity, SBN is Turing Complete [8].

---
**Algorithm 1** SBN $(a, b, c)$

---
**Require:  Input:** memory object $M$, addresses $a, b, c$
1:  $M[b] \leftarrow M[b] - M[a]$
2:  **if** $M[b] < 0$ **then**
3:      **go to** $c$
4:  **else**
5:      **go to next instruction**
6:  **end if**

---

SBN is closely related to SUBLEQ ("Subtrach and branch if less than or equal to zero"), differing only by the strict inequality instead of the inequality of SUBLEQ. This approach is fairly popular,

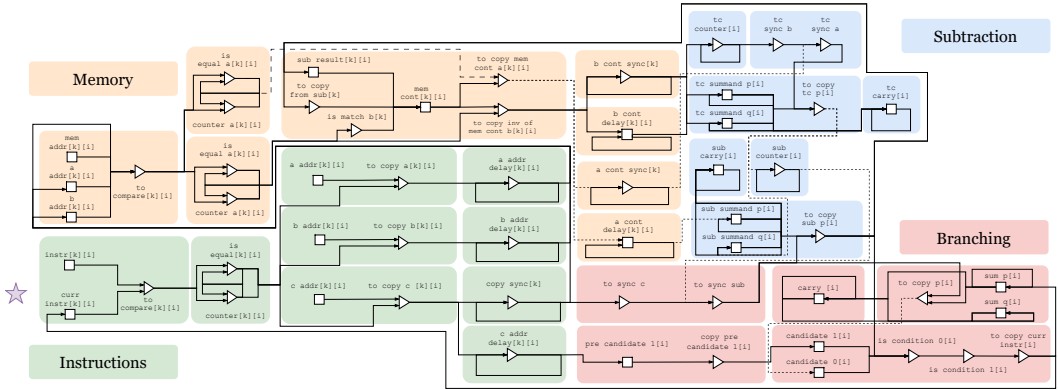

Figure 6: Overview of the simulation of SBN using the template-matching framework. Each rectangle in the diagram represents a block, with colors indicating specific block categories: Instructions, Memory, Branching, and Subtraction. Within each block are variables, depicted as shapes: squares represent data-holding variables, while triangles denote binary flags. Some blocks contain indexed variables. For simplicity, the sketch illustrates only one representative block for each unique index combination. Arrows between variables indicate interactions as defined by the instructions, representing the flow of data and control between variables and across different blocks. The star symbol on the left highlights blocks active at the start of an iteration. During this phase, each instruction is compared to the current one, triggering corresponding blocks on the right. An iterative bit-wise comparison of instruction addresses follows, and if a match occurs, copy flags for addresses $a$, $b$, and $c$ are triggered in the three adjacent right-hand blocks. The process continues according to the instructions detailed in the following sections.

with SUBLEQ being widely used in other works to demonstrate Turing Completeness in the context of Transformers [7, 1].

To simulate SBN within our framework, several auxiliary functions must be implemented in addition to those listed in Algorithm 1. These functions handle tasks such as retrieving an instruction from the list of instructions, accessing memory values from specified addresses, determining the next instruction based on the current one, and copying values between different fields, among other operations required by SBN.

The purpose of each function will become clear as their corresponding instructions are introduced. We will also explain the design choices behind them and provide the rationale for these decisions.

Before introducing the implementation details of our solution, we begin by outlining the structure we aim to simulate.

Our simulation involves two distinct objects: one for storing instructions and another for storing memory content. Conceptually, the instruction object can be viewed as a list of quadruples $(t, a, b, c)$, where $t$ is the address of the instruction (used for identification), and $a$, $b$, and $c$ are, respectively, two memory addresses and an instruction address. The triplet $(a, b, c)$ encodes the SBN instruction, as illustrated in Algorithm 1. Memory is structured as a list of pairs $(k, v)$, where $k$ denotes a memory address and $v$ its corresponding value.

For the input structure, following the framework outlined in Section 4, we divide the input into sets of blocks, each representing a large group of functions and variables. Each set of block is identified using the prefix specified in parentheses. The major blocks are organized as follows:

- **Instructions (I)**: Contains the list of instructions and their associated variables.
- **Memory (M)**: Contains the list of memory elements and their associated variables.
- **Branching (B)**: Handles the selection of the instruction based on the branching condition in line 2 of Algorithm 1, as well as the computation of the next instruction address.
- **Subtraction (D)**: Handles the subtraction of memory contents $a$ and $b$, as described in Algorithm 1.

Since both the instructions and memory objects contain multiple entries, we introduce additional notation. Let $\ell_I$ denote the number of instructions, which is assumed to be constant, as the number of instructions in an algorithm remains fixed, even though they may be executed repeatedly. Let $\ell_M$ denote the number of memory slots.

We define the number of bits required to address instructions as $n_I = \lceil \log_2 \ell_I \rceil$, and similarly, the number of bits required to address memory slots as $n_M = \lceil \log_2 \ell_M \rceil$. Additionally, each memory slot holds a value, whose bit-width we denote by $n_C$. An overview of our solution is illustrated in Figure 6.

In the following sections, we will derive the structure of the input $x$, using the quantities defined above to determine the number of blocks and the corresponding instructions.

**Blocks:** The total number of blocks is

$$\ell_M(5n_M + 3n_C + 3) + \ell_I(4n_I + n_M + 1) + 4n_I + 8n_C + 2 = \mathcal{O}(\ell_M \log \ell_M),$$

and the total length of the input $x$ is given by:

$$\ell_M(11n_M + 6n_C + 4) + \ell_I(8n_I + 4n_M + 1) + 11n_I + 12n_C + 4 = \mathcal{O}(\ell_M \log \ell_M).$$

The structure of the sets of blocks is defined as follows.

**Instructions blocks:** for $k \in [\ell_I]$, $i \in [n_I]$, $j \in [n_M]$:

- (I_to_compare[k][i], I_curr_instr[k][i], I_instr[k][i])
- (I_is_equal[k][i], I_counter[k][i])
- (I_a_addr[k][j], I_to_copy_a[k][j])
- (I_b_addr[k][j], I_to_copy_b[k][j])
- (I_c_addr[k][i], I_to_copy_c[k][i])
- (I_a_addr_delay[k][j])
- (I_b_addr_delay[k][j])
- (I_c_addr_delay[k][i])
- (I_copy_sync[k])

**Memory blocks:** for $k \in [\ell_M]$, $i \in [n_M]$, $j \in [n_C]$

- (M_to_compare[k][i], M_a_addr[k], M_b_addr[k][i], M_mem_addr[k][i])
- (M_is_equal_a[k][i], M_counter_a[k][i])
- (M_is_equal_b[k][i], M_counter_b[k][i])
- (M_mem_cont[k][j], M_to_copy_a_mem_cont[k][j], M_to_copy_inv_b_mem_cont[k][j], M_is_match_b[k][j], M_to_copy_from_sub[k][j], M_sub_result[k][j])
- (M_a_cont_delay[k][i])
- (M_a_cont_sync[k])
- (M_b_cont_delay[k][i])
- (M_b_cont_sync[k])

**Branching blocks:** for $i \in [n_I]$

- (B_sum_p[i], B_sum_q[i], B_to_copy_p[i])
- (B_carry[i])
- (B_to_sync_c, B_to_sync_sub)
- (B_pre_candidate_1[i], B_to_copy_candidate_1[i])
- (B_candidate_0[i], B_candidate_1[i], B_is_condition_0[i], B_is_condition_1[i] B_to_copy_curr_instr[i])

**Subtraction blocks:** for $i \in [n_C]$, $j \in [2n_C]$

- (D_tc_p[i], D_tc_q[i], D_to_copy_tc_p[i])
- (D_tc_sync_a, D_tc_sync_b)
- (D_tc_carry[i])
- (D_tc_counter[j])
- (D_sub_carry[i])
- (D_sub_p[i], D_sub_q[i], D_to_copy_sub_p[i])
- (D_sub_counter[j])

**Instructions**   Based on Algorithm 1 and the block structure outlined earlier, we now define the binary instructions for each process, grouped according to their corresponding block prefix. Considering all the instructions presented below, the total number of instructions is

$$\ell_M(13n_M + 7n_C + 1) + \ell_I(8n_I + 4n_M + 2) + 14n_C + 11n_I + 9 = \mathcal{O}(\ell_M \log \ell_M).$$

### Instructions: persist addresses (I)

The purpose of this function is to ensure that instruction addresses are not inadvertently deleted. Notably, we do not need to handle cases where the instruction address is zero, as any unmatched sample will naturally leave the corresponding entry as zero.

Additionally, note that there are no instructions dedicated to preserving the current instruction address `I_curr_instr` or the comparison flag `I_to_compare`. This omission is intentional, as both are temporary variables activated only in one execution stage and do not retain their values beyond that stage. Consequently, there is no need to explicitly preserve them.

```
INPUT: x[I_instr[k][i]] = 1 AND x[I_to_compare[k][i]] = 0 AND
       x[I_curr_instr[k][i]] = 0
OUTPUT: y[I_instr[k][i]] ← 1
Instruction count: ℓ_I · n_I
```

```
INPUT: x[I_a_addr[k][i]] = 1 AND x[I_to_copy_a[k][i]] = 0
OUTPUT: y[I_a_addr[k][i]] ← 1
Instruction count: ℓ_I · n_M
```

```
INPUT: x[I_b_addr[k][i]] = 1 AND x[I_to_copy_b[k][i]] = 0
OUTPUT: y[I_b_addr[k][i]] ← 1
Instruction count: ℓ_I · n_M
```

```
INPUT: x[I_c_addr[k][i]] = 1 AND x[I_to_copy_c[k][i]] = 0
OUTPUT: y[I_c_addr[k][i]] ← 1
Instruction count: ℓ_I · n_I
```

### Instructions: compare addresses (I)

In this function, the goal is to compare the current instruction address with the $k$-th instruction address, bit by bit. The result of this bitwise comparison is stored in a dedicated variable (`I_is_equal`), and a counter is activated to trigger a process that verifies whether all bits match. The set of instructions defined below handles both possible outcomes: when the addresses match and when they do not.

```
INPUT: x[I_to_compare[k][i]] = 1 AND x[I_curr_instr[k][i]] = x[I_instr[k][i]]
OUTPUT: (i = 1) y[I_instr[k][i]] ← x[I_instr[k][i]] AND y[I_is_equal[k][i]] ← 1
              AND y[I_counter[k][i]] ← 1
        (i > 1) y[I_instr[k][i]] ← x[I_instr[k][i]] AND y[I_is_equal[k][i]] ← 1
Instruction count: 2ℓ_I · n_I
```

```
INPUT: x[I_to_compare[k][i]] = 1 AND x[I_curr_instr[k][i]] ≠ x[I_instr[k][i]]
OUTPUT: (i = 1) y[I_instr[k][i]] ← x[I_instr[k][i]] AND y[I_counter[k][i]] ← 1
        (i > 1) y[I_instr[k][i]] ← x[I_instr[k][i]]
Instruction count: 2 · ℓ_I · n_I
```

**Instructions: check full address match and trigger copy (I)**

Following the previous stage, each of the comparison flags is evaluated iteratively using the counter variables. If all of them are equal to one, this indicates that the current instruction address exactly matches the $k$-th instruction address, which activates the copy flags. In the case where the `I_is_equal` variable is activated, but its corresponding `I_counter` has not yet been triggered, we preserve the value of `I_is_equal` using the second instruction.

```
INPUT: x[I_counter[k][i]] = 1 AND x[I_is_equal[k][i]] = 1
OUTPUT: (i < n_I) y[I_counter[k][i+1]] ← 1
        (i = n_I) y[I_to_copy_a[k][ALL]] ← 1 AND
                  y[I_to_copy_b[k][ALL]] ← 1 AND
                  y[I_to_copy_c[k][ALL]] ← 1
Instruction count: ℓ_I · n_I
```

```
INPUT: x[I_counter[k][i]] = 0 AND x[I_is_equal[k][i]] = 1
OUTPUT: y[I_is_equal[k][i]] ← 1
Instruction count: ℓ_I · n_I
```

**Instructions: copy address $a$, $b$ and $c$ (I)**

The goal of this function is to copy the matching $k$-th instruction triple $(a, b, c)$ into their respective blocks. This is achieved through two distinct sets of instructions. The first set defines the conditions under which copying should occur, while the second – denoted with the `_delay` and `_sync` suffix – is responsible for propagating the copy.

Strictly following the framework in Section 4, this second instruction set is technically unnecessary. One could, in principle, use the output from the final stage of the delayed process as the direct output of the first set. However, we adopt this two-stage implementation due to the nature of Neural Tangent Kernels (NTKs) and the challenge of managing write conflicts. If the alternative (single-stage) approach were used, the number of instructions writing to the same coordinates would increase with the number of instructions in the program, thereby leading to a proportional increase in write conflicts.

To mitigate this, we introduce a delay structure that propagates information sequentially across the instruction items. This design ensures that the number of write conflicts remains constant, regardless of the program size. While this approach incurs additional computational cost – in the form of more blocks and iterations – it does not hinder learnability, as discussed in Appendix C.

Additionally, since not all relevant variables are guaranteed to be set to 1, we introduce a supplementary variable for each, prefixed by `sync`. These `sync` variables function similarly to their `delay` counterparts but are always set to 1. This allows them to serve as a reliable synchronization mechanism across different processes.

```
INPUT: x[I_to_copy_a[k][i]] = 1 AND x[I_a_addr[k][i]] = 1
OUTPUT: y[I_a_addr_delay[k][i]] ← 1 AND y[I_a_addr[k][i]] ← 1
Instruction count: ℓ_I · n_M
```

```
INPUT: x[I_to_copy_b[k][i]] = 1 AND x[I_b_addr[k][i]] = 1
OUTPUT: y[I_b_addr_delay[k][i]] ← 1 AND y[I_b_addr[k][i]] ← 1
Instruction count: ℓ_I · n_M
```

```
INPUT: x[I_to_copy_c[k][i]] = 1 AND x[I_c_addr[k][i]] = 1
OUTPUT: (i = 1) y[I_c_addr[k][i]] ← 1 AND y[I_c_addr_delay[k][i]] ← 1 AND
                y[I_copy_sync[k]] ← 1
        (i > 1) y[I_c_addr[k][i]] ← 1 AND y[I_c_addr_delay[k][i]] ← 1
Instruction count: ℓ_I · n_I
```

```
INPUT: x[I_a_addr_delay[k][i]] = 1
OUTPUT: (k < ℓ_I) y[I_a_addr_delay[k+1][i]] ← 1
        (k = ℓ_I) y[M_a_addr[ALL][i]] ← 1
Instruction count: ℓ_I · n_M
```

```
INPUT: x[I_b_addr_delay[k][i]] = 1
OUTPUT: (k < ℓ_I) y[I_b_addr_delay[k+1][i]] ← 1
        (k = ℓ_I) y[M_b_addr][ALL][i] ← 1
Instruction count: ℓ_I · n_M
```

```
INPUT: x[I_c_addr_delay[k][i]] = 1
OUTPUT: (k < ℓ_I) y[I_c_addr_delay[k+1][i]] ← 1
        (k = ℓ_I) y[B_pre_candidate_1[i]] ← 1
Instruction count: ℓ_I · n_I
```

```
INPUT: x[I_copy_sync[k]] = 1
OUTPUT: (k < ℓ_I) y[I_copy_sync[k+1]] ← 1
        (k = ℓ_I) y[M_to_compare][ALL][ALL] ← 1 AND y[B_to_sync_c] ← 1
Instruction count: ℓ_I
```

**Instructions: persist addresses (M)**

As with the instruction block, the memory addresses must also be persisted. For the input block below, there is no need to handle scenarios where the other variables are equal to 1, as those cases are either already addressed during the comparison stage or do not arise during execution.

```
INPUT: x[M_to_compare[k][i]] = 0 AND x[M_mem_addr[k][i]] = 1 AND
x[M_a_addr[k][i]] = 0 AND x[M_b_addr[k][i]] = 0
OUTPUT: y[M_mem_addr[k][i]] ← 1
Instruction count: ℓ_M · n_M
```

**Instructions: compare addresses (M)**

In this stage, each memory address $k$ is compared to the addresses stored in the instruction fields $a$ and $b$. Note that the addresses from $a$ and $b$ have already been transmitted to their corresponding memory blocks. Below, we describe the behavior for different combinations of the values of M_a_addr, M_b_addr, and M_mem_addr.

```
INPUT: x[M_to_compare[k][i]] = 1 AND
       x[M_a_addr[k][i]] = x[M_mem_addr[k][i]] AND
       x[M_b_addr[k][i]] = x[M_mem_addr[k][i]]
OUTPUT: (i = 1) y[M_is_equal_a[k][i]] ← 1 AND y[M_is_equal_b[k][i]] ← 1 AND
               y[M_counter_a[k][i]] ← 1 AND y[M_counter_b[k][i]] ← 1 AND
               y[M_mem_addr[k][i]] ← x[M_mem_addr[k][i]]
        (i > 1) y[M_is_equal_a[k][i]] ← 1 AND y[M_is_equal_b[k][i]] ← 1 AND
               y[M_mem_addr[k][i]] ← x[M_mem_addr[k][i]]
Instruction count: 2ℓ_M · n_M
```

```
INPUT: x[M_to_compare[k][i]] = 1 AND
       x[M_a_addr[k][i]] ≠ x[M_mem_addr[k][i]] AND
       x[M_b_addr[k][i]] = x[M_mem_addr[k][i]]
OUTPUT: (i = 1) y[M_is_equal_b[k][i]] ← 1 AND y[M_counter_b[k][i]] ← 1 AND
               y[M_mem_addr[k][i]] ← x[M_mem_addr[k][i]]
        (i > 1) y[M_is_equal_b[k][i]] ← 1 AND
               y[M_mem_addr[k][i]] ← x[M_mem_addr[k][i]]
Instruction count: 2ℓ_M · n_M
```

```
INPUT: x[M_to_compare[k][i]] = 1 AND
       x[M_a_addr[k][i]] = x[M_mem_addr[k][i]] AND
       x[M_b_addr[k][i]] ≠ x[M_mem_addr[k][i]]
OUTPUT: (i = 1) y[M_is_equal_a[k][i]] ← 1 AND y[M_counter_a[k][i]] ← 1 AND
               y[M_mem_addr[k][i]] ← x[M_mem_addr[k][i]]
        (i > 1) y[M_is_equal_a[k][i]] ← 1 AND
               y[M_mem_addr[k][i]] ← x[M_mem_addr[k][i]]
Instruction count: 2ℓ_M · n_M
```

```
INPUT: x[M_to_compare[k][i]] = 1 AND
       x[M_a_addr[k][i]] ≠ x[M_mem_addr[k][i]] AND
       x[M_b_addr[k][i]] ≠ x[M_mem_addr[k][i]]
OUTPUT: x[M_mem_addr[k][i]] ← x[M_mem_addr[k][i]]
Instruction count: 2ℓ_M · n_M
```

**Instructions: check full address match and trigger copy (M)**

After computing bitwise equalities for each address and each address bit, the comparison flags for both $a$ and $b$ are checked iteratively. If all bits in a given comparison are equal to one, this indicates a match with the corresponding memory address, and the relevant processes are activated.

In the case of a match with address $a$, this triggers the process of copying the content of memory at address $a$ to the subtraction block. For a match with address $b$, a similar copying operation is triggered. However, instead of copying the bits directly, the inverse of each bit is copied. This serves to negate the content of $b$, as required by the subtraction logic, which will be explained in the following instructions.

Additionally, when a match is found in $b$, another flag is activated to indicate which memory slot corresponds to the current match. This flag is used later to update that memory slot with the result of the subtraction.

```
INPUT: x[M_counter_a][k][i] = 1 AND x[M_is_equal_a][k][i] = 1
OUTPUT: (i < n_M) y[M_counter_a][k][i+1] ← 1
        (i = n_M) y[M_to_copy_a_mem_cont][k][ALL] ← 1
Instruction count: ℓ_M · n_M
```

```
INPUT: x[M_counter_a][k][i] = 0 AND x[M_is_equal_a][k][i] = 1
OUTPUT: y[M_is_equal_a][k][i] ← 1
Instruction count: ℓ_M · n_M
```

```
INPUT: x[M_counter_b][k][i] = 1 AND x[M_is_equal_b][k][i] = 1
OUTPUT: (i < n_M) y[M_counter_b][k][i+1] ← 1
        (i = n_M) y[M_to_copy_inv_b_mem_cont][k][ALL] ← 1 AND
               y[M_is_match_b][k][ALL] ← 1
Instruction count: ℓ_M · n_M
```

```
INPUT: x[M_counter_b][k][i] = 0 AND x[M_is_equal_b][k][i] = 1
OUTPUT: y[M_is_equal_b][k][i] ← 1
Instruction count: ℓ_M · n_M
```

**Instructions: copy memory content (M)**

The following instructions cover all combinations of flags and memory content values. In this setting, we ensure that every valid combination of memory content and flag activation is captured. The transmission of memory information to the appropriate targets follows the same delay structure previously described, which helps manage write conflicts.

Importantly, the memory content is preserved in all cases, and the `M_is_match_b` flag is retained to indicate the corresponding memory slot for later updates to memory $b$.

Additionally, we must account for cases in which the memory content is null. This is necessary due to the flag `M_to_copy_inv_b_mem_cont`, which signals that the inverse of the memory content should be copied. As a result, its effect only arises when `M_mem_cont` is zero, whereas the effects of other flags are triggered when the memory content is non-zero.

```
INPUT: x[M_mem_cont][k][i] = 1 AND x[M_to_copy_a_mem_cont][k][i] = 1 AND
       x[M_to_copy_inv_b_mem_cont][k][i] = 1 AND x[M_is_match_b][k][i] = 1 AND
       x[M_to_copy_from_sub][k][i] = 0 AND x[M_sub_result][k][i] = 0
OUTPUT: (i = 1) y[M_mem_cont][k][i] ← 1 AND y[M_is_match_b][k][i] ← 1 AND
               y[M_a_cont_delay][k][i] ← 1 AND y[M_b_cont_sync][k] ← 1 AND
               y[M_a_cont_sync][k] ← 1
        (i > 1) y[M_mem_cont][k][i] ← 1 AND y[M_is_match_b][k][i] ← 1 AND
               y[M_a_cont_delay][k][i] ← 1
Instruction count: ℓ_M · n_C
```

```
INPUT: x[M_mem_cont][k][i] = 0 AND x[M_to_copy_a_mem_cont][k][i] = 1 AND
       x[M_to_copy_inv_b_mem_cont][k][i] = 1 AND x[M_is_match_b][k][i] = 1 AND
       x[M_to_copy_from_sub][k][i] = 0 AND x[M_sub_result][k][i] = 0
OUTPUT: (i = 1) y[M_is_match_b][k][i] ← 1 AND y[M_b_cont_delay][k][i] ← 1 AND
               y[M_b_cont_sync][k] ← 1 AND y[M_a_cont_sync][k] ← 1
        (i > 1) y[M_is_match_b][k][i] ← 1 AND y[M_b_cont_delay][k][i] ← 1
Instruction count: ℓ_M · n_C
```

```
INPUT: x[M_mem_cont][k][i] = 0 AND x[M_to_copy_a_mem_cont][k][i] = 0 AND
       x[M_to_copy_inv_b_mem_cont][k][i] = 1 AND x[M_is_match_b][k][i] = 1 AND
       x[M_to_copy_from_sub][k][i] = 0 AND x[M_sub_result][k][i] = 0
OUTPUT: (i = 1) y[M_is_match_b][k][i] ← 1 AND y[M_b_cont_delay][k][i] ← 1 AND
               y[M_b_cont_sync][k] ← 1
        (i > 1) y[M_is_match_b][k][i] ← 1 AND y[M_b_cont_delay][k][i] ← 1
Instruction count: ℓ_M · n_C
```

```
INPUT: x[M_mem_cont][k][i] = 1 AND x[M_to_copy_a_mem_cont][k][i] = 0 AND
       x[M_to_copy_inv_b_mem_cont][k][i] = 1 AND x[M_is_match_b][k][i] = 1 AND
       x[M_to_copy_from_sub][k][i] = 0 AND x[M_sub_result][k][i] = 0
OUTPUT: (i = 1) y[M_mem_cont][k][i] ← 1 AND y[M_is_match_b][k][i] ← 1 AND
               y[M_b_cont_sync][k] ← 1
        (i > 1) y[M_mem_cont][k][i] ← 1 AND y[M_is_match_b][k][i] ← 1
Instruction count: ℓ_M · n_C
```

```
INPUT: x[M_mem_cont][k][i] = 1 AND x[M_to_copy_a_mem_cont][k][i] = 1 AND
       x[M_to_copy_inv_b_mem_cont][k][i] = 0 AND x[M_is_match_b][k][i] = 0 AND
       x[M_to_copy_from_sub][k][i] = 0 AND x[M_sub_result][k][i] = 0
OUTPUT: y[M_mem_cont][k][i] ← 1 AND y[M_a_cont_delay][k][i] ← 1 AND
        y[M_a_cont_sync][k] ← 1
Instruction count: ℓ_M · n_C
```

```
INPUT: x[M_a_cont_delay[k][i]] = 1
OUTPUT: (k < ℓ_I) y[M_a_cont_delay[k+1][i]] ← 1
        (k = ℓ_M)  y[D_sub_p[i]] ← 1
Instruction count: ℓ_M · n_C
```

```
INPUT: x[M_b_cont_delay[k][i]] = 1
OUTPUT: (k < ℓ_I) y[M_b_cont_delay[k+1][i]] ← 1
        (k = ℓ_M) y[D_tc_p[i]] ← 1
Instruction count: ℓ_M · n_C
```

```
INPUT: x[M_a_cont_sync[k]] = 1
OUTPUT: (k < ℓ_M) y[M_a_cont_sync[k+1]] ← 1
        (k = ℓ_M)  y[D_tc_sync_a] ← 1
Instruction count: ℓ_M
```

```
INPUT: x[M_b_cont_sync[k]] = 1
OUTPUT: (k < ℓ_M) y[M_b_cont_sync[k+1]] ← 1
        (k = ℓ_M) y[D_tc_q[1]] ← 1 AND y[D_tc_counter[1]] ← 1
Instruction count: ℓ_M
```

## Instructions: negate memory content in $b$ (D)

This process takes place after the memory content has been effectively copied. The overall goal of the subtraction blocks is to compute the difference between the memory contents at addresses $a$ and $b$. To achieve this, we adopt a two-step procedure.

Before describing the procedure, we clarify that memory content is represented using two's complement encoding. Specifically, the least significant $n_C - 1$ bits represent the magnitude, while the most significant bit stores the sign.

Given this representation, subtraction is implemented by negating the content at address $b$ and then adding it to the content at address $a$.

In the previous step, the content from address $a$ was forwarded to a holding stage, awaiting the negated result of $b$. Meanwhile, we compute the two's complement negation of $b$ by first taking its bitwise inverse and then adding 1. The instructions below implement this stage, covering each operation involved in bitwise inversion, addition, carry propagation, and counter updates. These instructions follow the same structure described in Appendix B.2, but are limited to $n_C$ bits, meaning the final carry (MSB) is not propagated as an additional bit as it was done in Appendix B.2.

After $2n_C$ iterations, the addition is completed. The final counter triggers the copy flags, which forward the result to the same staging area as the content of $a$. In the second step, we sum these two values, yielding the desired result: $M[a] - M[b]$.

```
INPUT: x[D_tc_p[i]] = 1 AND x[D_tc_q[i]] = 0 AND x[D_to_copy_tc_p[i]] = 0
OUTPUT: y[D_tc_p[i]] ← 1
Instruction count: n_C
```

```
INPUT: x[D_tc_p[i]] = 0 AND x[D_tc_q[i]] = 1 AND x[D_to_copy_tc_p[i]] = 0
OUTPUT: y[D_tc_p[i]] ← 1
Instruction count: n_C
```

```
INPUT: x[D_tc_p[i]] = 1 AND x[D_tc_q[i]] = 1 AND x[D_to_copy_tc_p[i]] = 0
OUTPUT: y[D_tc_carry[i]] ← 1
Instruction count: n_C
```

```
INPUT: x[D_tc_carry[i]] = 1
OUTPUT: (i < n_C) y[D_tc_q[i+1]] ← 1
Instruction count: n_C - 1
```

```
INPUT: x[D_tc_counter[i]] = 1
OUTPUT: (i < 2n_C) y[D_tc_counter[i+1]] ← 1
        (i = 2n_C) y[D_tc_sync_b] ← 1
Instruction count: 2n_C
```

## Instructions: synchronize and trigger copy of negated content of $b$ (D)

In this operation, the copy is triggered only after confirming that both processes (negating the content of $b$ and copying the content of $a$) have been completed. This ensures no operation begins before all necessary inputs are available at their designated locations. If either of the two flags has not yet been activated, we preserve the current values until both are ready.

```
INPUT: x[D_tc_sync_a] = 1 AND x[D_tc_sync_b] = 1
OUTPUT: y[D_to_copy_tc_p][ALL] ← 1
Instruction count: 1
```

```
INPUT: x[D_tc_sync_a] = 1 AND x[D_tc_sync_b] = 0
OUTPUT: y[D_tc_sync_a] = 1
Instruction count: 1
```

```
INPUT: x[D_tc_sync_a] = 0 AND x[D_tc_sync_b] = 1
OUTPUT: y[D_tc_sync_b] = 1
Instruction count: 1
```

**Instructions: copy negated content of $b$ (D)**

Once the D_to_copy_tc_p flag is activated, the content of D_tc_p, which encodes the negation of the content of $b$, is copied to D_sub_q. This is the location where it will later be summed with the content of $a$.

In the following set of instructions, we do not include a case for when D_tc_q is 1, since after $2n_C$ iterations, D_tc_q is guaranteed to be zero.

```
INPUT: x[D_tc_p[i]] = 1 AND x[D_tc_q[i]] = 0 AND x[D_to_copy_tc_p[i]] = 1
OUTPUT: (i = 1) y[D_sub_q[i]] = 1 AND y[D_sub_counter][1] ← 1
        (i > 1) y[D_sub_q[i]] = 1
Instruction count: n_C
```

**Instructions: add the content of $a$ to the negated content of $b$ (D)**

Once the negated content of $b$ has been copied to the same stage as the content of $a$, the final subtraction result is obtained by performing a simple summation. The instructions below follow the same addition structure described in Appendix B.2. After completing $2n_C$ iterations, we trigger the sync flag to proceed to the next stage.

```
INPUT: x[D_sub_p[i]] = 1 AND x[D_sub_q[i]] = 0 AND x[D_to_copy_sub_p[i]] = 0
OUTPUT: y[D_sub_p[i]] ← 1
Instruction count: n_C
```

```
INPUT: x[D_sub_p[i]] = 0 AND x[D_sub_q[i]] = 1 AND x[D_to_copy_sub_p[i]] = 0
OUTPUT: y[D_sub_p[i]] ← 1
Instruction count: n_C
```

```
INPUT: x[D_sub_p[i]] = 1 AND x[D_sub_q[i]] = 1 AND x[D_to_copy_sub_p[i]] = 0
OUTPUT: y[D_sub_carry][i] ← 1
Instruction count: n_C
```

```
INPUT: x[D_sub_carry][i] = 1
OUTPUT: (i < n_C) y[D_sub_q][i+1] ← 1
Instruction count: n_C − 1
```

```
INPUT: x[D_sub_counter][i] = 1
OUTPUT: (i < 2n_C) y[D_sub_counter][i+1] ← 1
        (i = 2n_C) y[B_to_sync_sub] ← 1
Instruction count: 2n_C
```

**Instructions: copy subtraction result (D)**

At this stage, the flag D_to_copy_sub_p indicates that the content of D_sub_p, which holds the result of the subtraction, should be copied to all memory slots labeled as M_sub_result. This update is carried out using the matching flag M_is_match_b and the copy trigger M_to_copy_from_sub, as specified in Algorithm 1, in order to update the memory content at address $b$.

In the following set of instructions, we omit the case where D_sub_q is 1, since after $2n\_C$ iterations this variable should always be zero.

We also implement the condition from Algorithm 1 used to determine the next instruction. By checking the most significant bit (MSB) of D_sub_p, which represents the sign of the result, we decide the next step: if the MSB is 1, the result is negative and B_is_condition_1 is activated; otherwise, B_is_condition_0 is triggered.

```
INPUT: x[D_sub_p[i]] = 1 AND x[D_sub_q[i]] = 0 AND x[D_to_copy_sub_p[i]] = 1
OUTPUT: (i < n_C) y[M_sub_result[ALL][i]] ← 1
        (i = n_C) y[B_is_condition_1[ALL]] ← 1 AND y[M_sub_result[ALL][i]] ← 1 AND
                  y[M_to_copy_from_sub[ALL][ALL]] ← 1
Instruction count: n_C
```

```
INPUT: x[D_sub_p[i]] = 0 AND x[D_sub_q[i]] = 0 AND x[D_to_copy_sub_p[i]] = 1
OUTPUT: (i = n_C) y[B_is_condition_0[ALL]] ← 1 AND
                  y[M_to_copy_from_sub[ALL][ALL]] ← 1
Instruction count: 1
```

**Instructions: increment current instruction address (B)**

This set of instructions computes the address corresponding to the *else* condition in the branching logic of Algorithm 1. Specifically, it calculates the address of the *next instruction*, denoted by candidate_0, based on the current instruction address. The computation follows the same addition structure described in Appendix B.2.

By default, the initial vector $\hat{x}$ is configured such that the first current instruction is the all-zero vector, and the variable x[B_sum_p][1] is set to 1. This setup ensures that the algorithm always begins with the first instruction. After this initialization, the subsequent iterations proceed according to the logic defined by the instruction set. During the selection of the instruction address based on the branching condition, the chosen address is also copied to a scratchpad area, which is then used to compute $k + 1$. Once the $k + 1$ address is calculated, it is stored in the B_sum_p bits and retained until the appropriate copy flag is activated.

```
INPUT: x[B_sum_p[i]] = 1 AND x[B_sum_q[i]] = 0 AND x[B_to_copy_p[i]] = 0
OUTPUT: y[B_sum_p[i]] ← 1
Instruction count: n_I
```

```
INPUT: x[B_sum_p[i]] = 0 AND x[B_sum_q[i]] = 1 AND x[B_to_copy_p[i]] = 0
OUTPUT: y[B_sum_p[i]] ← 1
Instruction count: n_I
```

```
INPUT: x[B_sum_p[i]] = 1 AND x[B_sum_q[i]] = 1 AND x[B_to_copy_p[i]] = 0
OUTPUT: y[B_carry[i]] ← 1
Instruction count: n_I
```

```
INPUT: x[B_carry[i]] = 1
OUTPUT: (i < n_I) y[B_sum_q][i+1] ← 1
Instruction count: n_I − 1
```

**Instructions: copy next instruction address (B)**

Once the copying flag is activated, the contents of B_sum_p are copied to B_candidate_0. Simultaneously, we set B_sum_p[1] to 1 to ensure that it can increment the next instruction address during the next instruction update.

```
INPUT: x[B_sum_p[i]] = 1 AND x[B_sum_q[i]] = 0 AND x[B_to_copy_p[i]] = 1
OUTPUT: (i = 1) x[B_candidate_0[i]] = 1 AND x[B_sum_p[i]] = 1
        (i > 1) x[B_candidate_0[i]] = 1
Instruction count: n_I
```

```
INPUT: x[B_sum_p[i]] = 0 AND x[B_sum_q[i]] = 0 AND x[B_to_copy_p[i]] = 1
OUTPUT: (i = 1) x[B_sum_p[i]] = 1
Instruction count: 1
```

**Instructions: synchronize operations (B)**

In this operation, we synchronize the two independent phases: the subtraction and the retrieval of address $c$. Once both processes are complete, their results are simultaneously copied to the branching block. To ensure proper synchronization, we also include instructions that preserve the state of one flag if the other has not yet been activated.

```
INPUT: x[B_to_sync_c] = 1 AND x[B_to_sync_sub] = 1
OUTPUT: x[D_to_copy_sub_p[ALL]] ← 1 AND y[B_to_copy_p[ALL]] ← 1 AND
        y[B_to_copy_candidate_1[ALL]] ← 1
Instruction count: 1
```

```
INPUT: x[B_to_sync_c] = 1 AND x[B_to_sync_sub] = 0
OUTPUT: y[B_to_sync_c] ← 1
Instruction count: 1
```

```
INPUT: x[B_to_sync_c] = 0 AND x[B_to_sync_sub] = 1
OUTPUT: y[B_to_sync_sub] ← 1
Instruction count: 1
```

**Instructions: copy instruction address $c$ (B)**

Once synchronization is complete, we copy the value of B_candidate_1 from the staging area B_pre_candidate_1. The value in the staging area is preserved until the corresponding copy flag is activated, as detailed in the instructions below.

```
INPUT: x[B_pre_candidate_1[i]] = 1 AND x[B_to_copy_candidate_1[i]] = 1
OUTPUT: y[candidate_1[i]] ← 1
Instruction count: n_I
```

```
INPUT: x[B_pre_candidate_1[i]] = 1 AND x[B_to_copy_candidate_1[i]] = 0
OUTPUT: y[B_pre_candidate_1[i]] ← 1
Instruction count: n_I
```

**Instructions: update instruction address (B)**

Both candidate addresses are synchronously copied from their respective fields, along with the branching condition specified in Algorithm 1. Based on the activated condition, one of the candidates is selected. The chosen candidate is then copied into I_curr_instr, and I_to_compare is activated across all bits and instructions. Additionally, as previously noted, the selected instruction is also copied to B_sum_q to enable the computation of the next instruction address.

```
INPUT: x[B_is_condition_1][i] = 1 AND x[B_is_condition_0][i] = 0 AND
       x[B_candidate_1[i]] = 1 AND x[B_candidate_0[i]] = x[B_candidate_0[i]]
OUTPUT: (i < n_I) y[I_curr_instr][ALL][i] ← 1 AND y[B_sum_q[i]] ← 1
        (i = n_I) y[I_curr_instr][ALL][i] ← 1 AND
                 y[I_to_compare][ALL][ALL] ← 1 AND y[B_sum_q[i]] ← 1
Instruction count: 2n_I
```

```
INPUT: x[B_is_condition_1][i] = 1 AND x[B_is_condition_0][i] = 0 AND
       x[B_candidate_1[i]] = 0 AND x[B_candidate_0[i]] = x[B_candidate_0[i]]
OUTPUT: (i = n_I) y[I_to_compare][ALL][ALL] ← 1
Instruction count: 2
```

```
INPUT: x[B_is_condition_1][i] = 0 AND x[B_is_condition_0][i] = 1 AND
       x[B_candidate_0[i]] = 1 AND x[B_candidate_1[i]] = x[B_candidate_1[i]]
OUTPUT: (i < n_I) y[I_curr_instr][ALL][i] ← 1 AND y[B_sum_q[i]] ← 1
        (i = n_I) y[I_curr_instr][ALL][i] ← 1 AND
                 y[I_to_compare][ALL][ALL] ← 1 AND y[B_sum_q[i]] ← 1
Instruction count: 2n_I
```

```
INPUT: x[B_is_condition_1][i] = 0 AND x[B_is_condition_0][i] = 1 AND
       x[B_candidate_0[i]] = 0 AND x[B_candidate_1[i]] = x[B_candidate_1[i]]
OUTPUT: (i = n_I) y[I_to_compare][ALL][ALL] ← 1
Instruction count: 2
```

As required by the NTK learnability results, detailed in Appendix C, the following instructions exhibit a finite number of conflicts per coordinate. These conflicts are quantified by counting the number of outputs that write over the same coordinate. In this implementation, the maximum number of conflicts is 4. This occurs in the M_mem_addr bits, where memory values must be persisted under different combinations of control variables within the same block.

# C Proof of the NTK predictor behavior Theorem

In this section, we give the complete proof of Theorem 5.1. For convenience, we restate both the underlying assumption and the theorem below:

**Assumption 5.1.** For each test input $\hat{x}$ and for each position $i \in [k']$ such that the ground-truth output $f(\hat{x})$[10] has the $i$-th bit set, the number of training examples that do not match $\hat{x}$ and have the $i$-th bit set (which we call *unwanted correlations*) is less than the ratio $-w^1(\hat{x})/w^0(\hat{x})$.[11]

**Theorem 5.1** (NTK predictor behavior). *Consider an algorithmic problem cast as template-matching and encoded in a training set $(\mathcal{X}, \mathcal{Y}) \subseteq \mathbb{R}^{k'} \times \mathbb{R}^k$ as described in Section 5.1. Then, under Assumption 5.1, the mean of the limiting NTK distribution $\mu(\hat{x}) = \Theta(\hat{x}, \mathcal{X})\Theta(\mathcal{X}, \mathcal{X})^{-1}\mathcal{Y}$ for any test input $\hat{x} \in \mathbb{R}^{k'}$ contains sign-based information about the ground-truth output, namely for each coordinate of the output $i = 1, \ldots, k$, $\mu(\hat{x})_i \leq 0$ if the ground-truth bit at position $i$, $f(\hat{x})_i$, is set, and $\mu(\hat{x})_i > 0$ if the ground-truth bit at position $i$, $f(\hat{x})_i$, is not set.*

*Proof.* The proof begins by calculating the kernels $\Theta(\mathcal{X}, \mathcal{X}) \in \mathbb{R}^{kk' \times kk'}$ and $\Theta(\hat{x}, \mathcal{X}) \in \mathbb{R}^{k \times kk'}$. Using Equation (1)

$$\Theta := \Theta(\mathcal{X}, \mathcal{X}) = \left((d - c)I_{k'} + c\mathbf{1}\mathbf{1}^\top\right) \otimes I_k \in \mathbb{R}^{kk' \times kk'} \tag{10}$$

where

$$d = \frac{1}{k'} \quad \text{and} \quad c = \frac{1}{2\pi k'} \tag{11}$$

We can observe that since $d > c > 0$, $\Theta$ is positive definite. Indeed, since $\mathbf{1}\mathbf{1}^\top$ (the matrix of all ones) is positive semidefinite, $(d - c)I_{k'} + c\mathbf{1}\mathbf{1}^\top$ is (strictly) positive definite and so the Kronecker product with $I_k$ is also positive definite (see Appendix A). Similarly, the test NTK is given by

$$\Theta(\hat{x}, \mathcal{X}) = \mathbf{f}^\top \otimes I_k \in \mathbb{R}^{k \times kk'}$$

where for each $i = 1, 2, \ldots, k'$:

$$\mathbf{f}_i = \frac{\cos\hat{\theta}_i(\pi - \hat{\theta}_i) + \sin\hat{\theta}_i}{2k'\pi} + \hat{z}_i\frac{\pi - \hat{\theta}_i}{2k'\pi} \tag{12}$$

and

$$\hat{\theta}_i = \arccos(\hat{z}_i) \tag{13}$$

where $z_i$ is equal to 0 or $1/\sqrt{n_{\hat{x}}}$ depending on whether $\hat{x}$ matches the $i$-th training example. Since $z$ takes only two values, the resulting $\hat{\theta}_i$ takes two values $\hat{\theta}^0$ and $\hat{\theta}^1$ (corresponding to $\hat{x}_i = 0$ and $\hat{x}_i = 1/\sqrt{n_{\hat{x}}}$) and subsequently each $\mathbf{f}_i$ also takes two values $\mathbf{f}^0$ and $\mathbf{f}^1$. Finally, an application of Theorem A.1 gives

$$\Theta^{-1} = \left(\frac{1}{d - c}I_{k'} - \frac{c}{(d - c)(d - c + ck')}\mathbf{1}\mathbf{1}^\top\right) \otimes I_k$$

Substituting everything in $\Theta^{-1}\mathbf{f}$, we find that this takes the two values $w^0(\hat{x})$ and $w^1(\hat{x})$ as discussed in the main paper, namely:

$$(\Theta^{-1}\mathbf{f})_i = w^0(\hat{x}) = \frac{y^2 - \sqrt{y^2 - 1}y - 2\pi(y - 1) + 2y\sec^{-1}(y) - 1}{(2\pi - 1)(k + 2\pi - 1)}$$

if $\hat{x}$ does not match the $i$-th training sample and

$$(\Theta^{-1}\mathbf{f})_i = w^1(\hat{x}) = -\frac{\left(\sqrt{y^2 - 1} - y\right)\left(y^2 - k\right)}{(2\pi - 1)(k + 2\pi - 1)y} + \frac{\pi\left(k - y^2 + 2\sqrt{y^2 - 1} - 1\right)}{(2\pi - 1)(k + 2\pi - 1)y}$$
$$+ \frac{2\left(k - y^2 + 2\pi - 1\right)\csc^{-1}(y) - \sqrt{y^2 - 1} + 2\pi^2}{(2\pi - 1)(k + 2\pi - 1)y}$$

if $\hat{x}$ matches the $i$-th training sample, where $y = \sqrt{n_{\hat{x}}}$. Using a symbolic calculation system (Mathematica Inc. [12]) we can establish that for all possible values of $y$ (i.e $y = \sqrt{m}$ for $m = 1, 2, \ldots, b$), $w^0(\hat{x}) \leq 0$ and $w^1(\hat{x}) > 0$. The rest of the proof is exactly as given in the main paper. □

---

[10]There is a slight abuse of notation here when using $f(\hat{x})$ since $f$ does not operate on encoded inputs. Depending on the context, we may use $\hat{x}$ to denote both the pre-encoded and encoded test inputs.

[11]$w^0(\hat{x})$ is always non-positive and so the ratio is non-negative.

**Satisfying Assumption 5.1** To verify that Assumption 5.1 holds for a particular application we need to bound the number of unwanted correlations for each possible test input $\hat{\boldsymbol{x}}$. An easier way to verify that Assumption 5.1 holds is by studying the number of conflicts, as defined in Appendix B. Note that if the maximum number of conflicts is $c$, at most $c+1$ training examples can have a ground truth label of 1 at the same position. In particular, this shows that the number of unwanted correlations is at most $c$ for any test input $\hat{\boldsymbol{x}}$. In what follows we show how we leverage this observation to show that Assumption 5.1 is satisfied for all tasks discussed in the main paper:

- For the case of *binary permutations* of length $\ell$, we have $\ell$ template configuration and the input dimension is $k' = \ell$. Since we have no unwanted correlations, Assumption 5.1 is trivially satisfied.

- For the case of *binary addition* of two $\ell$-bit numbers, we have $2\ell$ template configurations and the input dimension is $k' = 4\ell$. We find symbolically that the ratio $-w^1(\hat{\boldsymbol{x}})/w^0(\hat{\boldsymbol{x}})$ is decreasing as a function of $n_{\hat{\boldsymbol{x}}}$ (for fixed $\ell$) and strictly greater than 1 for all $\ell \geq 1$ and $n_{\hat{\boldsymbol{x}}} \leq 2\ell$. Since the maximum number of conflicts is 1, Assumption 5.1 is satisfied.

- For the case of *binary multiplication* of two $\ell$-bit numbers, we have $11\ell$ template configurations and the input dimension is $k' = 20\ell + 2$. By adding extra training examples with corresponding ground truth labels of $\mathbf{0}$ that are never matched, we can augment the number of training examples to $k' = 21\ell$. We find symbolically that the ratio $-w^1(\hat{\boldsymbol{x}})/w^0(\hat{\boldsymbol{x}})$ is decreasing as a function of $n_{\hat{\boldsymbol{x}}}$ (for fixed $\ell$) and strictly greater than 2 for all $\ell \geq 1$ and $n_{\hat{\boldsymbol{x}}} \leq 7\ell + 1$.[12] Since the maximum number of conflicts is 2, Assumption 5.1 is satisfied.

- For the case of *SBN* with memory size $\ell_M = \ell$, we have $b = 5\ell(\log_2 \ell + c_1) + c_2 \log \ell + c_3$ template configurations and the input dimension is $k' = 13\ell(\log_2 \ell + c_4) + c_5 \log_2 \ell + c_6$, where $c_1, \ldots, c_6$ are positive constants based on other configurations of the algorithm being executed. In particular, notice that $b \leq C_1 \ell^2$ for some $C_1 > 0$. By a similar augmentation as before, we can achieve a dataset size of $k' = 5C_1\ell^2$. Again, we find symbolically that the ratio $-w^1(\hat{\boldsymbol{x}})/w^0(\hat{\boldsymbol{x}})$ is decreasing as a function of $n_{\hat{\boldsymbol{x}}}$ (for fixed $\ell$) and strictly greater than $5C_1/C_1 - 1 = 4$ for all $\ell \geq 1$ and $n_{\hat{\boldsymbol{x}}} \leq C_1\ell^2$ (and in particular for $n_{\hat{\boldsymbol{x}}} \leq b$). Since there are at most 4 unwanted correlations, Assumption 5.1 is satisfied. The previous argument can be repeated with $1 + \varepsilon$ for any $\varepsilon > 0$ in place of 2 at the exponent of $\ell$ and a different constant $C_\varepsilon > 0$.

The above is enough to conclude Remark 5.1.

## D Proof of the lemma for the order of the mean predictor and its variance

In this section, we provide the proof of Lemma 6.1. We do so by analyzing the mean and variance of the NTK predictor, expressing them as functions of the test vector $\hat{\boldsymbol{x}}$. Specifically, we characterize their dependence on the input length $k'$ and the number of matching blocks, $n_{\hat{\boldsymbol{x}}}$. We establish that the variance of the predictions scales as $\mathcal{O}\left(1/k'\right)$, while the mean coordinates exhibit different behaviors depending on whether the corresponding ground-truth bit is set and the relation between $\hat{\boldsymbol{x}}$ and $k'$. We begin by calculating the variance: a direct substitution on the formula for the variance $\Sigma(\hat{\boldsymbol{x}})$ of Equation (4) we find that the variance for a test input $\hat{\boldsymbol{x}}$ is given by

$$\Sigma(\hat{\boldsymbol{x}}) = \left( \frac{d}{2} + \mathbf{f}^\top A M A \mathbf{f} - 2\mathbf{f}^\top A \mathbf{g} \right) I_k$$

where

$$M = \left( \frac{d}{2} - c \right) I_{k'} + c \mathbf{1}\mathbf{1}^\top \qquad\qquad A = \frac{1}{d-c} I_{k'} - \frac{c}{(d-c)(d-c+ck')} \mathbf{1}\mathbf{1}^\top$$

and for each $i = 1, 2, \ldots, k'$:

$$\mathbf{g}_i = \frac{\cos \hat{\theta}_i (\pi - \hat{\theta}_i) + \sin \hat{\theta}_i}{2k'\pi}$$

---

[12] The maximum number of training examples that any test input can match is $7\ell + 1$. Refer to Appendix B for details.

The scalars $c$ and $d$, the vector $\mathbf{f}$, and the angles $\hat{\theta}_i$ are as defined in Equation (11), Equation (12) and Equation (13), respectively. This shows that the output coordinates are independent Gaussian random variables with the same covariance. Notice that whenever the test input is part of the training set, as expected, the variance vanishes.

**Preliminary quantities** To discharge notation and facilitate the subsequent proofs, we rewrite some of the already defined quantities and introduce some auxiliary quantities. Recall that:

$$d = \frac{1}{k'} \quad \text{and} \quad c = \frac{1}{2\pi k'}$$

We further introduce

$$d' = \frac{d}{2} = \frac{1}{2k'} \quad \text{and} \quad c' = c = \frac{1}{2\pi k'}$$

For a test vector $\hat{\boldsymbol{x}}$, we introduce the binary indicator $\mathbb{I}_i(\hat{\boldsymbol{x}})$ that indicates whether $\hat{\boldsymbol{x}}$ matches the $i$-th training example. With that, we rewrite:

$$\hat{\theta}_i = \begin{cases} \arccos\left(\frac{1}{\sqrt{n_{\hat{\boldsymbol{x}}}}}\right) & \text{if } \mathbb{I}_i(\hat{\boldsymbol{x}}) = 1 \\ \frac{\pi}{2} & \text{otherwise} \end{cases}$$

From this, we directly obtain the cosine and sine of $\hat{\theta}_i$:

$$\cos\hat{\theta}_i = \begin{cases} \frac{1}{\sqrt{n_{\hat{\boldsymbol{x}}}}} & \text{if } \mathbb{I}_i(\hat{\boldsymbol{x}}) = 1 \\ 0 & \text{otherwise} \end{cases} \qquad \sin\hat{\theta}_i = \begin{cases} \sqrt{1 - \frac{1}{n_{\hat{\boldsymbol{x}}}}} & \text{if } \mathbb{I}_i(\hat{\boldsymbol{x}}) = 1 \\ 1 & \text{otherwise} \end{cases}$$

Using these quantities, we rewrite:

$$\mathbf{g}_i = \begin{cases} \frac{1}{2\pi k\sqrt{n_{\hat{\boldsymbol{x}}}}}\left(\pi - \arccos\left(1/\sqrt{n_{\hat{\boldsymbol{x}}}}\right) + \sqrt{n_{\hat{\boldsymbol{x}}} - 1}\right) & \text{if } \mathbb{I}_i(\hat{\boldsymbol{x}}) = 1 \\ \frac{1}{2\pi k} & \text{otherwise} \end{cases}$$

and we get that Equation (12) is equal to

$$\mathbf{f}_i = \begin{cases} \mathbf{g}_i + \frac{\pi - \arccos\left(1/\sqrt{n_{\hat{\boldsymbol{x}}}}\right)}{2\pi k\sqrt{n_{\hat{\boldsymbol{x}}}}} & \text{if } \mathbb{I}_i(\hat{\boldsymbol{x}}) = 1 \\ \mathbf{g}_i & \text{otherwise} \end{cases}$$

## D.1 Computing the order of the variance

We begin by recalling the expression for the variance (substituting $d'$ and $c'$):

$$\Sigma(\hat{\boldsymbol{x}}) = \left(d' + \mathbf{f}^\top AMA\mathbf{f} - 2\mathbf{f}^\top A\mathbf{g}\right) I_k$$

where

$$M = (d' - c') I_{k'} + c'\mathbf{1}\mathbf{1}^\top \quad \text{and} \quad A = \frac{1}{d - c} I_{k'} - \frac{c}{(d-c)(d-c+ck')}\mathbf{1}\mathbf{1}^\top$$

We aim to show that the variance is bounded by $\mathcal{O}\left(1/k'\right)$. To facilitate this, we rewrite $\Sigma(\hat{\boldsymbol{x}})$ by expanding the matrix multiplication:

$$\Sigma(\hat{\boldsymbol{x}}) = d' I_k + \mathbf{f}^\top \left(AMA\mathbf{f} - 2A(\mathbf{f} - \mathbf{z})\right) I_k$$
$$= d' I_k + \mathbf{f}^\top \left(AMA\mathbf{f} - 2A\mathbf{f}\right) I_k + 2\mathbf{f}^\top A\mathbf{z}I_k$$

where we define $\mathbf{z} = \mathbf{f} - \mathbf{g}$ for notational simplicity. It is straightforward to observe that the first term, $d' I_k$, is bounded by $\mathcal{O}\left(1/k'\right)$. Thus, we focus our attention on the remaining two terms. For the second term, we begin by showing that $AMA - 2A$ has a maximum eigenvalue of $\mathcal{O}(1)$, and therefore:

$$\mathbf{f}^\top (AMA - 2A)\mathbf{f} \in \mathcal{O}\left(1/k'\right)$$

We begin by expressing $AMA$ in a more manageable form. A direct computation reveals:

$$AMA = u I_{k'} - v\mathbf{1}\mathbf{1}^\top$$

where

$$u = \frac{d' - c'}{(d - c)^2} \quad \text{and} \quad v = \frac{1}{k'}\left(\frac{d' - c' + k'c'}{(d - c + k'c)^2} - \frac{d' - c'}{(d - c)^2}\right)$$

Subtracting $2A$ from this expression gives:

$$AMA - 2A = (u - 2a)\, I_{k'} + (v + 2b)\, \mathbf{1}\mathbf{1}^\top$$

where

$$a = \frac{d - c}{(d - c)^2} \quad \text{and} \quad b = \frac{c}{(d - c + ck')\,(d - c)}$$

This matrix has two unique eigenvalues: $u - 2a$ (with multiplicity $k' - 1$) and $u - 2a + (v + 2b)k'$. We will show that the largest eigenvalue is $\mathcal{O}(1)$. To this end, we evaluate these two quantities, starting with $u - 2a$:

$$u - 2a = \frac{d' - c' - 2(d - c)}{(d - c)^2}$$

$$= -\frac{2\pi(3\pi - 1)k'}{(2\pi - 1)^2}$$

which is negative. For the second eigenvalue, we find:

$$u - 2a + (v + 2b)k' = -\frac{2\pi(3\pi - 1)k'}{(2\pi - 1)^2} + \frac{2\pi k'\,(\pi + k' - 1)}{(2\pi + k' - 1)^2} - \frac{2\pi k'(\pi - 1)}{(2\pi - 1)^2}$$

$$+ \frac{4\pi k'}{(2\pi - 1)\,(2\pi + k' - 1)} = \left(\frac{4\pi}{2\pi + k' - 1} + \frac{\pi + k' - 1}{(2\pi + k' - 1)^2}\right)k'$$

which is positive and clearly $\mathcal{O}(1)$. With this result, we can bound the multiplication by the norms of its components. Since $\lambda_{\max}(AMA - 2A) \in \mathcal{O}(1)$ and $\|\mathbf{f}\|^2 \in \mathcal{O}\left(1/k'\right)$ (since each $\mathbf{f}_i \in \mathcal{O}\left(1/k'\right)$), we conclude:

$$\mathbf{f}^\top(AMA - 2A)\mathbf{f} \in \mathcal{O}\left(1/k'\right)$$

We now turn to the third term, $2\mathbf{f}^\top A\mathbf{z}$. Expressing $\mathbf{z}$ component-wise we have:

$$\mathbf{z}_i = \begin{cases} \frac{\pi - \arccos(1/\sqrt{n_{\hat{\boldsymbol{x}}}})}{2\pi k' \sqrt{n_{\hat{\boldsymbol{x}}}}} & \text{if } \mathbb{I}_i(\hat{\boldsymbol{x}}) \neq 0 \\ 0 & \text{otherwise} \end{cases}$$

We then decompose the product $\mathbf{f}^\top A\mathbf{z}$ as:

$$\mathbf{f}^\top A\mathbf{z} = a\mathbf{f}^\top \mathbf{z} - b(\mathbf{f}^\top \mathbf{1})(\mathbf{1}^\top \mathbf{z}) \tag{14}$$

where

$$a = \frac{1}{d - c} \quad \text{and} \quad b = \frac{2\pi k'}{(2\pi - 1)(2\pi + k' - 1)}$$

The first term $a\mathbf{f}^\top \mathbf{z}$ can be expressed as:

$$a\mathbf{f}^\top \mathbf{z} = \frac{1}{(2\pi - 1)(2\pi k')^2}\left(2\pi - 2\arccos\left(1/\sqrt{n_{\hat{\boldsymbol{x}}}}\right) - \sqrt{n_{\hat{\boldsymbol{x}}} - 1}\right)\left(\pi - 2\arccos\left(1/\sqrt{n_{\hat{\boldsymbol{x}}}}\right)\right)$$

which is positive and $\mathcal{O}\left(1/k'\right)$. For the second term $b(\mathbf{f}^\top \mathbf{1})(\mathbf{1}^\top \mathbf{z})$, we start by expressing the individual quantities $\mathbf{f}^\top \mathbf{1}$ and $\mathbf{1}^\top \mathbf{z}$:

$$\mathbf{f}^\top \mathbf{1} = \frac{\sqrt{n_{\hat{\boldsymbol{x}}}}}{\pi k'}\left(\pi - \arccos\left(1/\sqrt{n_{\hat{\boldsymbol{x}}}}\right)\right) + \frac{\sqrt{n_{\hat{\boldsymbol{x}}}}\sqrt{n_{\hat{\boldsymbol{x}}} - 1}}{2\pi k'} + \frac{k' - n_{\hat{\boldsymbol{x}}}}{2\pi k'}$$

and

$$\mathbf{1}^\top \mathbf{z} = \frac{\sqrt{n_{\hat{\boldsymbol{x}}}}}{2\pi k'}\left(\pi - \arccos\left(1/\sqrt{n_{\hat{\boldsymbol{x}}}}\right)\right)$$

Therefore, the entire term can be expressed as:

$$b(\mathbf{f}^\top \mathbf{1})(\mathbf{1}^\top \mathbf{z}) = \frac{1}{2\pi k'(2\pi - 1)(2\pi - 1 + k')}\Big( 2n_{\hat{\boldsymbol{x}}}\left(\pi - \arccos\left(1/\sqrt{n_{\hat{\boldsymbol{x}}}}\right)\right)^2$$

$$+ n_{\hat{\boldsymbol{x}}}\sqrt{n_{\hat{\boldsymbol{x}}} - 1}\left(\pi - \arccos\left(1/\sqrt{n_{\hat{\boldsymbol{x}}}}\right)\right)$$

$$+ (k' - n_{\hat{\boldsymbol{x}}})\sqrt{n_{\hat{\boldsymbol{x}}}}\left(\pi - \arccos\left(1/\sqrt{n_{\hat{\boldsymbol{x}}}}\right)\right) \Big)$$

which is positive and $\mathcal{O}\left(1/k'\right)$, therefore, the subtraction of Equation (14) (which is equal to $\mathbf{f}^\top A\mathbf{z}$) is $\mathcal{O}\left(1/k'\right)$. Combining the bounds on all three terms, we conclude that:

$$\Sigma(\hat{\boldsymbol{x}}) \in \mathcal{O}\left(1/k'\right)I_k$$

completing the proof of the first part of Lemma 6.1.

### D.2 Computing the order of the mean

Recall from Equation (7) that each coordinate of the NTK predictor mean is given as a weighted sum of $w^1 \equiv w^1(\hat{\boldsymbol{x}})$ and $w^0 \equiv w^0(\hat{\boldsymbol{x}})$, where $w^1 = (\Theta^{-1}\mathbf{f})_i$ for all coordinates $i \in [k]$ such that $\hat{\boldsymbol{x}}$ matches the $i$-th training example, and $w^0(= (\Theta^{-1}\mathbf{f})_i$ for all coordinates $i \in [k]$ such that $\hat{\boldsymbol{x}}_i$ does not match the $i$-th training example. In particular, the template-matching mechanism of Equation (6) shows that whenever the $i$-th ground-truth bit of $\hat{\boldsymbol{x}}$, $f(\hat{\boldsymbol{x}})_i$, is set, the NTK predictor satisfies:

$$\mu(\hat{\boldsymbol{x}})_i = w^1 + |\mathcal{I}_-^1(\hat{\boldsymbol{x}})| \cdot w^0 \tag{15}$$

where $\mathcal{I}_-^1(\hat{\boldsymbol{x}})$ denotes the indices of training examples that do not match $\hat{\boldsymbol{x}}$ and have their $i$-th ground-truth output bit set. Similarly, whenever $f(\hat{\boldsymbol{x}})_i = 0$, the NTK predictor satisfies:

$$\mu(\hat{\boldsymbol{x}})_i = |\mathcal{I}_-^1(\hat{\boldsymbol{x}})| \cdot w^0 \tag{16}$$

Since $|\mathcal{I}_-^1(\hat{\boldsymbol{x}})|$ is bounded by the maximum number of conflicts (as defined in Appendix B) which is constant for all four tasks (i.e. it doesn't scale with $k'$), the asymptotic order of $\mu(\hat{\boldsymbol{x}})$ is determined solely by the asymptotic orders of $w^0$ and $w^1$. We can thus turn our attention to

$$\Theta^{-1}\mathbf{f} = \left( \frac{1}{d-c}I_{k'} - \frac{c}{(d-c)(d-c+ck')}\mathbf{1}\mathbf{1}^\top \right)\mathbf{f}$$

which we decompose into two terms:

$$\Theta^{-1}\mathbf{f} = \frac{\mathbf{f}}{d-c} - \frac{c\mathbf{1}\mathbf{1}^\top\mathbf{f}}{(d-c)(d-c+ck)} \tag{17}$$

For the first term in Equation (17), we obtain for each $i = 1, 2, \ldots, k'$:

$$\frac{\mathbf{f}_i}{d-c} = \begin{cases} \frac{1}{2\pi-1}\left( \frac{\pi-\arccos(1/\sqrt{n_{\hat{\boldsymbol{x}}}})}{\sqrt{n_{\hat{\boldsymbol{x}}}}} + \sqrt{1 - \frac{1}{\sqrt{n_{\hat{\boldsymbol{x}}}}}} \right) & \text{if } \mathbb{I}_i(\hat{\boldsymbol{x}}) = 1 \\ \frac{1}{2\pi-1} & \text{otherwise} \end{cases}$$

The second term is a constant vector with coefficient:

$$\frac{c\mathbf{1}\mathbf{1}^\top\mathbf{f}}{(d-c)(d-c+ck')} = \frac{1}{(2\pi-1)(2\pi-1+k')}\left( \frac{n_{\hat{\boldsymbol{x}}}\left(\pi - \arccos\left(1/\sqrt{n_{\hat{\boldsymbol{x}}}}\right)\right)}{2\sqrt{n_{\hat{\boldsymbol{x}}}}} \right.$$

$$\left. + n_{\hat{\boldsymbol{x}}}\sqrt{1 - \frac{1}{n_{\hat{\boldsymbol{x}}}}} + k' - n_{\hat{\boldsymbol{x}}} \right)$$

We now evaluate each case separately.

**Case 1:** For $w^0(\hat{\boldsymbol{x}})$, that is, when $\mathbb{I}_i(\hat{\boldsymbol{x}}) = 0$, we have:

$$w^0(\hat{\boldsymbol{x}}) = \frac{1}{2\pi-1}\left( 1 - \frac{k-n_{\hat{\boldsymbol{x}}}}{2\pi+k'-1} - \frac{2\pi k' n_{\hat{\boldsymbol{x}}}}{(2\pi+k'-1)\pi k'\sqrt{n_{\hat{\boldsymbol{x}}}}}\left(\pi - \arccos\left(1/\sqrt{n_{\hat{\boldsymbol{x}}}}\right)\right) \right)$$

$$- \frac{1}{2\pi-1}\left( \frac{2\pi k' n_{\hat{\boldsymbol{x}}}}{(2\pi+k'-1)\,2\pi k'}\sqrt{1 - \frac{1}{\sqrt{n_{\hat{\boldsymbol{x}}}}}} \right) \tag{18}$$

The absolute value of Equation (18) behaves like $\Theta\left(\sqrt{n_{\hat{\boldsymbol{x}}}}/k'\right)$, and setting $n_{\hat{\boldsymbol{x}}}$ to different regimes yields the bounds:

$$
|w^0(\hat{\boldsymbol{x}})| \in
\begin{cases}
\Theta\left(1/k'\right) & \text{if } n_{\hat{\boldsymbol{x}}} \text{ is constant} \\
\Theta\left(\sqrt{n_{\hat{\boldsymbol{x}}}}/k'\right) & \text{if } n_{\hat{\boldsymbol{x}}} \text{ is non-constant} \\
& \text{and sublinear in } k' \\
\Theta\left(1/\sqrt{k'}\right) & \text{if } n_{\hat{\boldsymbol{x}}} = ck' \text{ for } c \in (0,1]
\end{cases}
$$

**Case 2:** For $w^1(\hat{\boldsymbol{x}})$, that is, when $\mathbb{I}_i(\hat{\boldsymbol{x}}) = 1$, we have:

$$
\begin{aligned}
w^1(\hat{\boldsymbol{x}}) = {}& \frac{2\pi - 1}{2\pi + k' - 1}\left(\frac{2}{(2\pi - 1)\sqrt{n_{\hat{\boldsymbol{x}}}}}\left(\pi - \arccos\left(1/\sqrt{n_{\hat{\boldsymbol{x}}}}\right)\right) + \frac{1}{2\pi - 1}\sqrt{1 - \frac{1}{\sqrt{n_{\hat{\boldsymbol{x}}}}}}\right) \\
& + \frac{k' - n_{\hat{\boldsymbol{x}}}}{2\pi + k' - 1}\left(\frac{2}{(2\pi - 1)\sqrt{n_{\hat{\boldsymbol{x}}}}}\left(\pi - \arccos\left(1/\sqrt{n_{\hat{\boldsymbol{x}}}}\right)\right)\right. \\
& \left. + \frac{1}{2\pi - 1}\sqrt{1 - \frac{1}{\sqrt{n_{\hat{\boldsymbol{x}}}}}} - \frac{1}{2\pi - 1}\right)
\end{aligned}
\tag{19}
$$

When setting $n_{\hat{\boldsymbol{x}}} = k'$, we note that the second term becomes zero and the first term becomes $\Theta\left(1/k'\right)$. Alternatively, the first term in Equation (19) is $\Theta\left(1/k'\right)$ and the second term is $\Theta\left(1/\sqrt{n_{\hat{\boldsymbol{x}}}}\right)$, implying $\mu(\hat{\boldsymbol{x}})_1 \in \Theta\left(1/\sqrt{n_{\hat{\boldsymbol{x}}}}\right)$. Setting $n_{\hat{\boldsymbol{x}}}$ to different regimes yields the bounds established in Lemma 6.1, namely:

$$
|w^1(\hat{\boldsymbol{x}})| \in
\begin{cases}
\Theta\left(1/\sqrt{n_{\hat{\boldsymbol{x}}}}\right) & \text{if } n_{\hat{\boldsymbol{x}}} \text{ is non-constant} \\
& \text{and sublinear in } k', \\
\Theta\left(1/\sqrt{k'}\right) & \text{if } n_{\hat{\boldsymbol{x}}} = ck' \text{ for } c \in (0,1) \\
\Theta\left(1/k'\right) & \text{if } n_{\hat{\boldsymbol{x}}} = k'
\end{cases}
$$

Substituting the derived orders in Equation (16) and Equation (15) yields the orders of Lemma 6.1, completing the proof of the second part of Lemma 6.1.

**Remark D.1.** *The conclusion of Lemma 6.1 holds for any task such that the cardinality of $\mathcal{I}_-^1(\hat{\boldsymbol{x}})$ does not scale with $k'$ for any test input $\hat{\boldsymbol{x}}$. In particular, it holds for tasks such that the maximum number of conflicts is bounded by a constant that does not scale with the number of bits. To interpret the last condition visually, consider $\mathcal{Y}$ as in Figure 3. We require each column of $\mathcal{Y}$ to have a sum bounded by a constant which does not scale with the number of bits (and hence the input dimension $k'$). For example, that constant for addition is equal to $2$.*

## E   Training Experiments on Permutation

We present two experiments that empirically validate our theoretical findings for the algorithmic task of permutation. We chose this task because it requires significantly fewer models to achieve reasonable results compared to more complex tasks such as binary addition and multiplication, which exhibit substantially higher ensemble complexity. Addressing those tasks would demand large-scale computational infrastructure, which is currently beyond our available resources. Nonetheless, our present results still offer strong empirical support for our theoretical conclusions.

For this task, we train a two-layer fully connected feed-forward network with a hidden layer of width 50,000 only using standard basis vectors and optimized using full-batch gradient descent. Our goal is to learn some random permutation on normalized binary inputs of length $k$. For testing, we evaluate the performance on 1000 random vectors with $n_{\hat{\boldsymbol{x}}} = 2$ nonzero entries. To match the assumptions of our theoretical analysis, we initialize the weights exactly as given in Section 3.

**1. Number of models to achieve $90\%$ accuracy.** Our first experiment examines how many independently trained models need to be averaged to achieve a testing accuracy of 90% as a function of the input length $k$.

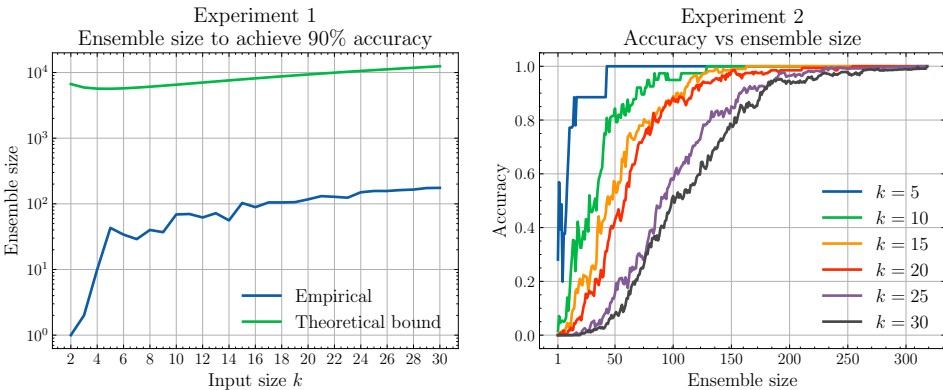

Figure 7: Experiment 1 (left) presents the required number of models (ensemble size) to achieve 90% accuracy as a function of input size $k$, compared to the theoretical bound from Equation (8) with the appropriate $\delta$. Experiment 2 (right) presents accuracy as a function of ensemble size for different input sizes $k$.

**2. Accuracy vs ensemble size.** Our second experiment fixes $k \in \{5, 10, 15, 20, 25, 30\}$ and examines how the post-rounding accuracy varies as the ensemble size increases.

In Figure 7, the left plot presents the results of Experiment 1 compared to the bound established in Equation (8) for the appropriate choice of $\delta$ to guarantee 90% accuracy for all test inputs simultaneously. We observe that the empirical ensemble size remains below the theoretical bound, and that both curves follow a similar pattern as a function of $k$. For Experiment 2, the results on the right plot of Figure 7 illustrate the convergence to perfect accuracy as a function of the ensemble size for different input sizes $k$. As shown, larger input sizes require more models to achieve perfect accuracy, but with a sufficient number of models, all instances eventually reach perfect accuracy.

