# OpenReview forum: "Learning to Add, Multiply, and Execute Algorithmic Instructions Exactly with Neural Networks"
_NeurIPS.cc/2025/Conference — NeurIPS 2025 poster_

### Official Review · Reviewer_eFhn · 2025-06-04

**Clarity:** 3
**Significance:** 3
**Originality:** 3
**Rating:** 5
**Confidence:** 2

**Summary:**

This paper gives theoretical results showing that an MLP with 1 hidden layer can learn various bitwise algorithms, using the NTK infinite-width framing. The general idea is to partition the neurons into "blocks", each of which functions effectively as a memory slot, and to assume training examples which illustrate how each block transforms independently. Provided that those training examples are provided, and that there are not inter-block correlations, they show that various algorithms can be learned, given step-by-step examples of each block transformation, and with an acceptable sample complexity (logarithmic in the input space, so polynomial in the number of bits). They support their claims with numerical simulations.

**Questions:**

I am very confused by the claim that the results extend to arbitrary algorithms simply because SBN is Turing complete. It seems that your setup assumes a fixed amount of memory. How can it be Turing complete in that case?

**Ethical Concerns:**

["NO or VERY MINOR ethics concerns only"]

**Final Justification:**

The authors were upfront in the discussion about the memory limitations issues / extrapolation to bigger inputs and said they would discuss in revision.

**Limitations:**

yes

**Paper Formatting Concerns:**

no concerns. beautiful figures!

**Quality:**

3

**Strengths And Weaknesses:**

Strengths:
1. It's important to understand the computational abilities of neural networks, and while much past work has addressed what neural networks can represent, this work further addresses what is actually learnable
2. The work appears rigorous, although I am not an expert in the relevant theoretical areas
3. I found the work to be original and creative.

Weaknesses:
1. The empirical validation is explained well in the appendix, but given that the NeurIPS crowd has a more empirical leaning, it would be wise to migrate more of this discussion to the main text (for example, I wanted to know how many hidden nodes you use in order to approximate the "infinite width" limit)
2. I'm not so sure about the Turing-completeness claim, see question

---

> ### Author Rebuttal · Authors · 2025-07-30
>
> We sincerely appreciate the reviewer’s time and effort in evaluating our manuscript. We are grateful for their recognition of the significance of our work. Below, we address their thoughtful and constructive comments.
>
> **Weaknesses:**
>
> > W1: The empirical validation is explained well in the appendix, but given that the NeurIPS crowd has a more empirical leaning, it would be wise to migrate more of this discussion to the main text (for example, I wanted to know how many hidden nodes you use in order to approximate the "infinite width" limit)
>
> We appreciate the reviewer’s suggestion, which makes good sense. Due to the submission page limit, we had to restrict the main text and place some of the empirical details in the appendix. We agree that including more of this discussion in the main text would benefit readers, and we plan to do so in future versions of the paper.
>
> **Questions:**
>
> > Q1: I am very confused by the claim that the results extend to arbitrary algorithms simply because SBN is Turing complete. It seems that your setup assumes a fixed amount of memory. How can it be Turing complete in that case?
>
> The reviewer raises a valid and important point, which we address below.
>
> In any realization of a computational model, memory must be bounded in some way, just as physical computers have finite memory. However, the critical aspect of our exact learning framework is that it allows for arbitrarily large memory. That is, if a particular computation requires more memory, the input size and the number of instructions need to scale (in a trivial way) to accommodate the extra memory requirement. Importantly, our exact learning result demonstrates this remains learnable, even as the memory requirements grow.

---

> > ### Comment · Reviewer_eFhn · 2025-08-04
> >
> > Thank you for agreeing to migrate more of the empirical results to the main text.
> >
> > I hope that the revision can also clarify the memory limitations. I'm confused by whether increasing the memory limit would also increase the number of parameters. Your response suggests to me that is the case, in which case you would have to do more learning in order to fit those parameters. A hallmark of successful algorithm learning is the ability to train on short simple examples and then extrapolate to harder test examples that require more memory. Would this be possible for your setup?

---

> > > ### Author Response · Authors · 2025-08-04
> > >
> > > We thank the reviewer for participating actively in the discussion period.
> > >
> > > The reviewer asks whether our bounded-memory framework, trained on short, simple instances, can extrapolate to much longer inputs (i.e., length generalize) without a prohibitive parameter blow-up as the memory size $\ell$ increases. First, we assume that by “number of parameters”, the reviewer refers to the input dimension. We note that in our SBN result, the input dimension scales as $O(\ell \log \ell)$. This means that the input dimension grows linearithmically with the memory length, which is efficient. We would also like to note that, under the bounded-memory assumption, our framework allows full algorithmic execution by training on only logarithmically many short and simple examples. However, changing the memory size and, consequently, the input dimension of the MLP, requires retraining.
> > >
> > > This naturally raises the question of whether architectures designed for variable-length inputs, such as RNNs or Transformers, could achieve length generalization within our framework. Provable, assumption-free (arbitrary) length generalization remains an open problem for exact learning. To the best of our knowledge, no other framework, statistical or otherwise, provably achieves such a strong generalization result.
> > >
> > > Having said that, it can still be achieved under some conditions in our framework for certain applications. For example, we are currently working on extending the results of the present work to graph tasks using GNNs on bounded-degree graphs. The use of a GNN architecture, combined with the bounded-degree assumption, supports length generalization in terms of the number of nodes in the graph, despite the overall memory of the neural network being bounded. In that case, the extent of length generalization is limited only by the finite precision used to represent positional encodings of the nodes, and not by the memory size of the neural network.
> > >
> > > A thorough theoretical investigation of these questions is an exciting direction for future work and will likely yield valuable insights into the exact learning capabilities of neural architectures.
> > >
> > > We would be happy to include a discussion of such limitations in the revised version of our paper.

---

### Official Review · Reviewer_MLRy · 2025-06-24

**Clarity:** 2
**Significance:** 4
**Originality:** 3
**Rating:** 5
**Confidence:** 3

**Summary:**

The paper proves that NTK-trained networks can learn algorithms solvable by local template matching.

**Questions:**

- In the caption for Fig. 2, what do the authors mean by "although the method is iterative, this example completes in one step"? Also, can the authors clarify if template matching can only provide such iterative solutions?
- Could the authors replot the two curves in each subplot of Fig. 5 such that they coincide? Also I'd like to know what the constant factors are in each case.
- How does Turing completeness allow Assumption 5.1 to hold?
- Since the experiments were done with finite-sized networks, do the authors have any insight for asymptotics on $n_h$ - the width of the network?

**Ethical Concerns:**

["NO or VERY MINOR ethics concerns only"]

**Final Justification:**

The paper addresses my concerns, and I still recommend this paper to be accepted.

**Limitations:**

There are no important limitations that were not highlighted by the authors.

**Quality:**

3

**Strengths And Weaknesses:**

Strengths:
- The paper presents a rigorous proof that NTK-trained neural networks can learn template matching with high probability.
- The authors give several examples of algorithms that can be solved via template matching, and hence, prove exact learnability of such algorithms with NTK-trained models.

Weaknesses:
- The paper suffers from some notational ambiguity. For example, in Section 4 the block configuration is represented by $\mathbf{x}$, but the dependence on the block $B_i$ is suppressed.

---

> ### Author Rebuttal · Authors · 2025-07-30
>
> We sincerely thank the reviewer for their careful evaluation of our manuscript. We appreciate their recognition of the contributions of our work. Below, we respond to their insightful comments and questions.
>
> **Weaknesses:**
>
> > W1: The paper suffers from some notational ambiguity. For example, in Section 4 the block configuration is represented by x, but the dependence on the block B_i is suppressed
>
> We thank the reviewer for highlighting this notational ambiguity. To improve clarity and readability, we omitted the dependence on the block $B_i$, as it can be inferred from the context. We will make this explicit in future versions of the manuscript.
>
> **Questions:**
>
> > Q1: In the caption for Fig. 2, what do the authors mean by "although the method is iterative, this example completes in one step"?
>
> Under the template matching framework, our solution to the addition task typically requires at most $2\ell$ iterations, where $\ell$ refers to the number of bits. This represents a worst-case upper bound. However, some input samples may require fewer iterations. For instance, if no carry values need to be propagated to subsequent bits, the computation can be completed in a single pass. This is illustrated in Figure 2, which is why it can be completed in just one step, despite the iterative nature of the method. We apologize for any confusion and will revise the caption to improve clarity.
>
> > Q2: Can the authors clarify if template matching can only provide such iterative solutions?
>
> The principle of template matching is highly versatile and, in some cases, can be applied in configurations that eliminate the need for iteration. The implications of this depend on the complexity of the problem. For instance, in simple tasks such as binary permutations (discussed in the paper), a single pass is sufficient. However, for more complex tasks, such as binary addition, removing the iterative component requires significantly more samples (potentially exponential in the input length) to achieve single-pass execution. Thus, the framework’s iterative nature is key to maintaining a small dataset size.
>
> Additionally, the number of iterations is directly tied to the algorithm encoded in the instructions. In some cases, an alternative algorithm might reduce the required number of iterations.
>
> Finally, as previously noted, the iteration count may also vary depending on the samples being processed. If the focus is limited to a specific, simpler subset of samples, such as "easy" cases in binary addition, the iteration requirement might be relaxed or even eliminated.
>
> > Q3: Could the authors replot the two curves in each subplot of Fig. 5 such that they coincide? Also I'd like to know what the constant factors are in each case.
>
> As requested, we plotted each theoretical estimate of the sample complexity, scaled by a factor $c$ to align the empirical and theoretical curves. Unfortunately, due to this year’s NeurIPS guidelines, authors are not permitted to include plots or external links in the response. For the three tasks shown in Fig. 5, we find that the curves have a mean relative error of less than 1% when the theoretical estimates are scaled by $c = 4.9$ for permutation, $c = 6.7$ for addition, and $c = 6.0$ for multiplication. We plan to include these results, along with the corresponding plots, in future versions of the paper.
>
> > Q4: How does Turing completeness allow Assumption 5.1 to hold?
>
> We assume the reviewer is puzzled by the fact that, despite the rigidity of Assumption 5.1, our exact learning results generalize to computable functions. While Assumption 5.1 is rigid, it is important to clarify that our goal in learning SBN instructions is to learn the computations of a universal computer itself. To this end, we carefully designed a set of instructions such that the number of conflicts in each coordinate of the output (corresponding to the ratio in Assumption 5.1) remains constant. Once trained, the resulting model functions as a learned machine that can accept inputs encoding arbitrary computations, much like machine code. Therefore, to execute any computable function, the neural network must be provided with the corresponding machine code as input. In contrast, if we were to attempt to learn an arbitrary computable function directly (without an intermediate computational model like SBN acting as an interpreter) it is possible that, for certain computations, no set of instructions could satisfy Assumption 5.1.
>
> > Q5: Since the experiments were done with finite-sized networks, do the authors have any insight for asymptotics on - the width of the network?
>
> The reviewer asks an important question regarding the convergence error as a function of the network's width. This pertains to the field of finite-width corrections in NTK theory, a topic with a rich literature of its own (see, for example, the survey [1]).
>
> In this study, we intentionally focused on the infinite-width regime. This choice was strategic, as this setting provides the cleanest possible framework to convey our central message: that, under specific assumptions, neural networks are capable of learning to execute algorithmic instructions exactly. By eliminating the complexities of finite-width effects (e.g., dealing with higher-order kernels as defined in Section 3 of [1]), we can provide a clear proof of this principle.
>
> A thorough theoretical analysis of the finite-width case would be a significant research effort in its own right, one that we believe would warrant a separate, dedicated paper. We opted to prioritize the clarity and impact of our core message, but we agree that understanding these corrections might be a promising next step in this research direction.
>
> [1] Golikov et al. (2022). Neural Tangent Kernel: A Survey

---

> > ### Comment · Reviewer_MLRy · 2025-08-04
> >
> > Thanks for addressing my concerns. I'll keep my original score.

---

> > > ### Author Response · Authors · 2025-08-04
> > >
> > > We sincerely thank the reviewer once more for their careful evaluation and for recognizing the value of our work.

---

### Official Review · Reviewer_1CbE · 2025-06-27

**Clarity:** 3
**Significance:** 3
**Originality:** 3
**Rating:** 4
**Confidence:** 2

**Summary:**

This paper investigates whether neural networks, specifically two-layer fully connected ReLU networks in the infinite-width limit (analyzed via the Neural Tangent Kernel or NTK), can learn to execute algorithmic instructions exactly. The authors demonstrate that by structuring training data into orthogonal bit-level templates and analyzing training dynamics in the NTK regime, such models can provably learn four core algorithmic tasks: binary permutations, addition, multiplication, and Subtract and Branch if Negative (SBN) instructions—the latter being Turing-complete. They prove that exact learning is possible using a logarithmic number of training examples, and provide ensemble complexity bounds to ensure high-probability correctness after rounding.

**Questions:**

1. Can the authors clarify how to obtain the orthogonal training set in practical scenarios? This assumption is central to the theoretical results but may not hold for real-world training datasets. Could approximate orthogonality suffice?


2. Your setup leads to exact learning under NTK dynamics with small datasets. How do your findings relate to grokking phenomena, where models trained with standard datasets undergo sudden generalization?


3. The analysis assumes infinite width and NTK regime. Have you tested the fidelity of your predictions with finite-width networks trained with SGD?


4. Assumption 5.1 is central to the correctness of the NTK predictor. How sensitive are your results to this assumption? Can the authors provide more intuition (beyond the bound of 2 unwanted correlations) or empirical plots showing how performance degrades as it is violated?

**Ethical Concerns:**

["NO or VERY MINOR ethics concerns only"]

**Final Justification:**

Rebuttal address my concerns.

**Limitations:**

Yes. The authors explicitly discuss their limitations in Section 7, including the assumptions of orthogonal template training and direct instruction access. They also propose several promising directions to relax these constraints.

**Quality:**

3

**Strengths And Weaknesses:**

## Strengths

1. Provides rigorous theoretical results showing exact learnability of discrete algorithmic tasks.

2. Uses NTK formalism effectively to derive precise guarantees, going beyond simulation-based evidence.

3. Covers Turing-complete instruction sets (via SBN), showing generality.

4. Results are supported by illustrative examples, ensemble bounds, and practical template encodings.

## Weakness

1. Main result theorem 5.1 requires assumption 5.1. Also the paper assume orthogonal training data and explicit template access, which may limit applicability in more realistic, data-driven settings.

2. Does not address noisy or partial supervision, which is relevant for LLMs and real-world execution.

3. Empirical validation is limited; most claims rely on infinite-width limits and concentration results. Practical significance may be limited unless these theoretical insights can be extended to finite-width or real-world training settings.

---

> ### Author Rebuttal · Authors · 2025-07-30
>
> We thank the reviewer for their time and feedback. Below, we address the reviewer’s concerns and answer their questions, one by one.
>
> **Weaknesses:**
>
> > W1: Main result theorem 5.1 requires assumption 5.1.
>
> We assume that the reviewer perceives Assumption 5.1 as a limitation. However, it is satisfied by (Turing-complete) SBN instructions, allowing for a very expressive model. This means that for any given program, there exists an SBN-based implementation that satisfies Assumption 5.1 and can thus be learned by our framework. Relaxing this assumption, or showing it is necessary for exact learnability, would be an exciting direction for future research. Finally, even for in-distribution learning (a much more relaxed learning framework than exact learning), gradient-based end-to-end learning of algorithmic execution is provably hard without further assumptions such as curriculum learning or CoT-type supervision.
>
> > Also the paper assume orthogonal training data and explicit template access [...]
>
> We emphasize that our results concern exact learning: we make no assumptions about data distributions, and the output must be exactly correct with high probability, and not just approximately correct, as in PAC learning. Exact learning is a relatively new theoretical framework [1], and our work offers a good first step toward its understanding.
>
> In the exact learning setting, the central question of our work is: can neural networks perform a fundamental form of algorithmic computation? Assuming perfect knowledge of each elementary step, we examine whether a network trained via gradient descent can execute the full algorithm and produce the exact bit-level output. To our knowledge, this is the first provable, positive result addressing this question. Going beyond orthogonal training data and explicit template access is certainly an exciting future work direction.
>
> > W2: Does not address noisy or partial supervision, which is relevant for LLMs and real-world execution.
>
> While we understand the reviewer’s perspective, we would like to reiterate that our work specifically examines the ability of neural networks to learn and execute algorithms exactly.  To the best of our knowledge, we provide the first proof that neural networks can execute algorithms both exactly and correctly when trained solely on the necessary instructions. We agree that developing a theoretical framework for exact learning to explain the real-world execution behavior of modern architectures, such as LLMs, is a highly compelling goal. However, we believe that achieving such a framework will require significant foundational progress, particularly in establishing exact learning results. In our view, tackling simpler, more analyzable settings is a necessary and valuable step in this direction. We see our work as a concrete and meaningful contribution toward that long-term objective (cf. [1]).
>
> > W3: Empirical validation is limited; most claims rely on infinite-width limits and concentration results. Practical significance may be limited unless these theoretical insights can be extended to finite-width or real-world training settings.
>
> Our theoretical claims are supported by empirical results presented in Appendix E, where we train ensembles of two-layer fully connected neural networks (hidden dimension 50,000) on the binary permutation task. We observe that accuracy improves with ensemble size, eventually reaching 100%. Notably, the ensemble size required to achieve 90% accuracy remains well below our theoretical infinite-width ensemble complexity bound.
>
> To further address the reviewer’s interest, we conducted a similar experiment for the binary addition task. Using the same network architecture, we trained ensembles on instruction sequences for addition (as shown in Section 4.1 and Appendix B.2). Below, we provide a table showing the ensemble sizes required to reach 90% accuracy, alongside the theoretical bounds (from Figure 5) for bit-lengths $\ell \in \\{3,4,5\\}$. In all cases, the required ensemble size remains below the theoretical threshold.
>
> | **Number of bits**                 | 3    | 4    | 5    |
> |------------------------------------|------|------|------|
> | **Empirical ensemble size to reach 90% acc.**       | 870  | 990  |   1090  |
> | **Numerical ensemble complexity bound** | 4003 | 6691 | 9926 |
>
> **Questions:**
>
> > Q1: Can the authors clarify how to obtain the orthogonal training set in practical scenarios? This assumption is central to the theoretical results but may not hold for real-world training datasets. Could approximate orthogonality suffice?
>
> We detail the dataset construction in Section 5.1. In brief, we map the set of inputs to an orthonormal set of vectors (e.g., the standard basis) and feed this into the neural network instead. The procedure is straightforward: we iterate through the original inputs and assign each one a unique standard basis vector. This pre-processing step is instrumental as it minimizes the number of unwanted correlations during inference, effectively enabling sign-based recovery of the ground-truth output. This process is efficient since the neural network’s input dimension is equal to the size of the dataset, which, in turn, scales linearly in the number of bits.
>
> The question of approximate orthogonality is a very interesting one from a theoretical standpoint and is one of the points discussed in Section 7: “Limitations and future work”. Intuitively, the strictness of the exact learning requirement requires strong control over unwanted correlations. From a modeling standpoint, an orthonormal training dataset is the most natural choice to achieve this. Yet, it would be interesting to see if, for specific tasks, this could be relaxed (perhaps at the expense of requiring a more sophisticated rounding function or a larger ensemble size) or if this is impossible altogether.
>
> > Q2: Your setup leads to exact learning under NTK dynamics with small datasets. How do your findings relate to grokking phenomena, where models trained with standard datasets undergo sudden generalization?
>
> We appreciate the reviewer’s insightful question. Below, we elaborate on this point.
> Grokking represents another learning regime in neural networks, which is very interesting but not covered by the present work. In this context, [2] offers useful insights by contrasting learning regimes and suggesting that grokking may exhibit higher data efficiency. However, grokking remains a developing area, and to our knowledge, no unifying theoretical framework has yet been proposed that parallels the NTK formulation for lazy training.
>
> We also wish to clarify our notion of the training set, which differs from the standard input-output format used in both grokking and NTK studies. For example, in the binary addition task, our training samples are binary instructions rather than input-output pairs (e.g., summands and their sum). This formulation eliminates sample correlations and enables exact execution over all valid inputs. While unconventional, it allows us to prove exact learnability, a result that, to our knowledge, has not been previously established.
>
> > Q3: The analysis assumes infinite width and NTK regime. Have you tested the fidelity of your predictions with finite-width networks trained with SGD?
>
> Our theoretical claims have been empirically tested through experiments with ensembles of finite-width networks, as shown in Appendix E. A summary of these results is also provided in our response to “W3” above. We recognize the importance of this aspect and are happy to move portions of Appendix E into the main paper in the revised version.
>
> > Q4: Assumption 5.1 is central to the correctness of the NTK predictor. How sensitive are your results to this assumption? [...]
>
> We thank the reviewer for this insightful question. At a high level, Assumption 5.1 ensures that unwanted correlations with incorrect samples do not dominate the signal from correct instructions in the training data. As Assumption 5.1 is violated, the NTK predictor’s behavior can deteriorate, and the results of Theorem 5.1 may not hold.
>
> To empirically validate its necessity under our learning framework, we construct a toy dataset that incrementally increases the number of unwanted correlations. The dataset consists of inputs with 100 blocks (each of size 1) and 1-dimensional binary outputs. Each input is a standard basis vector, with output value equal to 1 for $p$ randomly chosen data points, and output 0 for the remaining $100-p$ data points.
>
> We construct an unseen test input by summing one (out of $p$) training input with output 1 and several inputs with output 0. By the template-matching principle, the correct output should be 1. Since only one of the $p$ positive-output inputs is matched, the remaining $p-1$ introduce unwanted correlations, which we refer to as conflicts.
>
> The table below shows how the NTK predictor’s mean changes with increasing conflicts ($p-1$), reflecting violations of Assumption 5.1. As the number of conflicts grows, the mean drops and eventually reaches a point where the predictor’s sign fails to encode the correct output.
>
> | **Number of conflicts** | 1       | 20      | 40      | 60      | 80      | 90      |
> |-------------------------|---------|---------|---------|---------|---------|---------|
> | **NTK predictor mean**  | 0.36646 | 0.27907 | 0.18708 | 0.09508 | 0.00309 | -0.0429 |
>
>
> ---
>
> Once again, we sincerely thank the reviewer for taking the time to read our manuscript. We welcome any further questions or comments and hope that our clarifications are sufficient for the reviewer to consider raising their score.
>
>
> [1] György et al. (2025). Beyond Statistical Learning: Exact Learning Is Essential for General Intelligence
>
> [2] Kumar et al. (2024). Grokking as the Transition from Lazy to Rich Training Dynamics. ICLR 2024

---

> > ### Comment · Reviewer_1CbE · 2025-08-05
> >
> > Thanks for addressing my concerns. I have raised my score

---

> > > ### Author Response · Authors · 2025-08-05
> > >
> > > We sincerely thank the reviewer for reading our rebuttal and raising their score.

---

### Official Review · Reviewer_8X6V · 2025-07-02

**Clarity:** 2
**Significance:** 2
**Originality:** 3
**Rating:** 4
**Confidence:** 2

**Summary:**

This paper considers the learnability of a particular class of boolean functions (expressed as a disjunction of simpler blocks) by two-layer ReLU networks with large width. Appealing to NTK theory, they suggest that the NTK limit should converge to approximate functions of this form satisfying a particular margin-like assumption (Assumption 5.1).

**Questions:**

Rather than saying logarithmic sample complexity (implicitly, in the size of the number of possible inputs, which is confusing), say sample complexity polynomial in the input size (i.e., in the number of bits needed to code an input)

What is the convergence error of the large-width network to the NTK limit as a function of the hidden dimension?

**Ethical Concerns:**

["NO or VERY MINOR ethics concerns only"]

**Final Justification:**

I thank the reviewers for their thorough response. My biggest issue was being able to verify the correctness of the core results (i.e., Theorem 5.1). I find the reviewers' elaboration helpful and see the high-level idea better, though I still would not say I can rigorously validate all details (though this could be due to a lack of familiarity with the area). I have therefore raised my score.

**Paper Formatting Concerns:**

nit: h.c vs. h.c. on page 3

**Quality:**

3

**Strengths And Weaknesses:**

## Strengths

1. The paper addresses the problem of learning primitive discrete operations with feedforward networks that, when composed, could be computationally powerful. This is tangentially related to methods like chain of thought, and theoretical understanding of the learning difficulty could be interesting and important.
2. Section 3 on NTK (an existing result) and Section 4 on templates seem rigorous and correct

## Weaknesses

1. The problem considered here could be much better motivated. The authors confusingly allude to Turing completeness but are not particularly careful or clear. The point is that iteratively composing a sequence of boolean functions of the form considered here is quite expressive. You consider the learning problem for one step in this chain - not the end-to-end learning problem. It’s important to clarify this both so that readers do not get the wrong impression and to make it clear what the motivation of the paper is in the first place.
2. Related to the motivation, the authors should cite and engage with significant existing work on on the expressivity and learnability of iteratively applying simple neural functions to compute more complex ones (e.g., attaining Turing completeness via chain of thought):
    1. https://arxiv.org/abs/2309.06979
    2. https://arxiv.org/abs/2107.13163
3. I am not able to verify the correctness of Sections 5 and 6, which constitute the core new results. In particular, I do not even understand Assumption 5.1, which is central the message here. Why is w0 non-positive if it’s defined as the proportion of training samples that do not match a block? Shouldn’t a proportion always be positive? This part of the paper should be clarified so that it is easier to understand and verify the results.

---

> ### Author Rebuttal · Authors · 2025-07-30
>
> We sincerely appreciate the reviewer’s time in evaluating our manuscript. Below, we address their insightful comments in order of appearance:
>
> **W1. Paper motivation**
>
> We appreciate the reviewer’s suggestion regarding the presentation of our findings. To clarify, we want to ensure our intent is not misinterpreted.
> > The point is that iteratively composing a sequence of boolean functions of the form considered here is quite expressive.
>
> We agree that this composition is indeed expressive, and our SBN exact learning result aims to quantify that expressiveness. Specifically, we learn a set of Boolean operations that, in a loop, enable emulation of a general-purpose computer (SBN), which is a common approach in the literature (see [3,4,5]) to demonstrate expressiveness.
> > You consider the learning problem for one step in this chain - not the end-to-end learning problem.
>
> We believe this is a potential misunderstanding. Our goal was not to learn a universal computer end-to-end. We build upon existing literature that uses the standard looping mechanism to demonstrate the expressivity of neural networks (see [3,4,5]). While most of these works are constructive, our goal was to *exactly learn* the instructions of the looped SBN computer. In our setup, each training data point corresponds to a single instruction that encodes part of an algorithm. As noted in the paper, iterative execution enables composition across steps, allowing full algorithmic behavior. Only the Boolean permutation task requires a single step. All other computations, SBN, addition, and multiplication, require multiple iterations, as specified. We welcome the reviewer to highlight any unclear passages so that we can revise them for clarity.
>
> **W2. Related work**
>
> > Related to the motivation, the authors should cite and engage with significant existing work [...]
>
> We sincerely thank the reviewer for providing such valuable references, which help to better position our contributions. We discuss their distinctions in more detail below.
> As mentioned above, our learnability results apply to the setting where the goal is to emulate a general-purpose computer. In our case, we aim to learn the instructions of the universal computer itself. Once trained, the model can accept inputs that encode arbitrary computations, analogous to machine code. This contrasts with the reviewer-cited work [1], where the learning target is a specific Turing-computable function, rather than the computational model that interprets such functions.
> Both settings are quite reasonable, and as such, we will incorporate the works cited by the reviewer into the related work section. As noted above, we also commit to revising relevant parts of the manuscript to clarify the distinctions between our Turing-completeness results and those presented in [1,2].
> With this clarification, we hope our intended positioning of the paper is clearer. That said, in light of the reviewer’s references, we acknowledge that certain passages may admit multiple interpretations. To this end, we will revise such passages (e.g., L10–11, L200–201), outlining how such Turing Completeness results differ from works such as [1,2]. We sincerely thank the reviewer for highlighting these points and for sharing these insightful references. We also welcome further suggestions on how to more effectively present these ideas.
>
> **W3. Clarification on results**
> > I am not able to verify the correctness of Sections 5 and 6, which constitute the core new results. [...] This part of the paper should be clarified [...].
>
> We appreciate the reviewer’s concern regarding the clarity of our theoretical results. We take this opportunity to elaborate on the meaning of Assumption 5.1, which we intend to incorporate into our paper to improve clarity. The reviewer is correct that a proportion must be positive. However, $w_0$ and $w_1$ *are not proportions*; they are scalar weights derived from the structure of the NTK matrices, as described below.
>
> As described in Section 5.2, due to input data orthogonalization, the NTK predictor
>
> $$\mu(\hat{x}) = \Theta(\hat{x}, X)\Theta(X, X)^{-1}\mathcal{Y}$$
>
> can be rewritten as:
>
> $$\mu(\hat{x}) = w_1 \sum_{i \in S(\hat{x})} y_i + w_0 \sum_{j \notin S(\hat{x})} y_j$$
>
> In other words, the NTK predictor is a weighted sum of training output examples, where the weights depend on whether the corresponding training input matches the test input $\hat{x}$, whose matching indexes are indicated by $S(\hat{x})$. Hence, $w_1$ and $w_0$ are not proportions but rather sample-level weights, with $w_1 > 0$ and $w_0 \leq 0$.
>
> For each coordinate of the NTK predictor, our goal is to have the same sign as the correct output: non-positive if the true label is 0 and positive otherwise.
> To illustrate this, consider the case where we want a particular coordinate of the predictor to be equal to 1. In this case, the weighted sum must be positive. This requires that the negative contribution from output examples with non-zero entries in that coordinate does not outweigh the positive contribution from the correct output example. This is the intuition formalized in Assumption 5.1, which requires that the number of “unwanted correlations” remains below a certain ratio. This condition ensures that the sign of each coordinate reflects the desired classification behavior.
> We hope this clarification helps convey the intuition behind Assumption 5.1 and improves the overall readability of the theoretical discussion.
>
> **Questions:**
>
> > Q1: Rather than saying logarithmic sample complexity [...], say sample complexity polynomial in the input size (i.e., in the number of bits needed to code an input)
>
> We thank the reviewer for this constructive feedback. This is an excellent point about clarity, and we will revise our wording accordingly.
> Our choice of “logarithmic in the number of all possible inputs” was motivated by two factors. First, we aimed to highlight the significant efficiency of our framework. For an $\ell$-bit input, the space of all possible inputs is exponential ($2^{\ell}$), while our required training data scales polynomially with $\ell$. Phrasing this as logarithmic in the total number of inputs was our way of underscoring this exponential reduction.
> Second, we were cautious about using the term “sample complexity”, as it typically implies a statistical PAC-learning setting with random data drawn according to some underlying distribution. Our analysis, in contrast, uses a fixed, deterministically constructed dataset and provides guarantees for any possible test input.
> That said, we agree that expressing the dataset size in terms of the input's bit length is more conventional and ultimately clearer for most readers. We will update the text to incorporate this change.
> > Q2: What is the convergence error of the large-width network to the NTK limit as a function of the hidden dimension?
>
> The reviewer asks an important question regarding the convergence error as a function of the network's width. This pertains to the field of finite-width corrections in NTK theory, a topic with a rich literature of its own (see survey [6], for example).
> In this study, we intentionally focused on the infinite-width regime. This choice was strategic, as this setting provides a clear framework to convey our central message: that, under specific assumptions, neural networks are capable of learning to execute algorithmic instructions exactly. By removing the complexities of finite-width effects (e.g., dealing with higher-order kernels as defined in Section 3 of [6]), we can offer a clear proof of this principle.
> A thorough theoretical analysis of the finite-width case would be a significant research effort in its own right, one that we believe would warrant a separate, dedicated paper. We opted to prioritize the clarity and impact of our core message, but we agree that understanding these corrections might be a promising next step in this research direction.
> That being said, the reviewer might find the experiments in Appendix E interesting, as they provide an empirical probe into the behavior of finite-width networks. There, we train an ensemble of two-layer fully connected feedforward neural networks with a hidden dimension of 50,000 on the binary permutation task. Briefly, we find that as the ensemble size grows, the accuracy increases and eventually reaches 100%. Furthermore, the ensemble size required to reach 90% accuracy is always well below our theoretical, infinite-width ensemble complexity upper bound. To further satisfy the reviewer’s curiosity, we also conducted a similar experiment for the addition task. Namely, we trained ensembles of two-layer fully connected feedforward neural networks with a hidden dimension of 50,000 on the instructions for binary addition, as described in Section 4.1 and Appendix B.2. Below, we present a table containing the ensemble sizes required to reach 90% accuracy as well as the numerical bound (from Figure 5) for bit-lengths $\ell\in \\{3,4,5\\}$. Observe that the ensemble size still falls below the bound.
> | **Number of bits**                 | 3    | 4    | 5    |
> |------------------------------------|------|------|------|
> | **Empirical ensemble size to reach 90% acc.**       | 870  | 990  |  1090 |
> | **Numerical ensemble complexity bound** | 4003 | 6691 | 9926 |
>
> ---
> Once again, we thank the reviewer and welcome any further comments. We hope the clarifications are sufficient for them to consider raising their score.
>
> [1] Malach (2024). Auto-Regressive Next-Token Predictors are Universal Learners
>
> [2] Wei et al. (2023). Statistically Meaningful Approximation: a Case Study on Approximating Turing Machines with Transformers
>
> [3] Siegelmann et al. (1992). On the computational power of neural nets
>
> [4] Perez et al (2021). Attention is Turing-Complete
>
> [5] Giannou et al. (2023). Looped Transformers as Programmable Computers
>
> [6] Golikov et al. (2022). Neural Tangent Kernel: A Survey

---

> > ### Comment · Reviewer_8X6V · 2025-08-04
> >
> > I thank the reviewers for their thorough response. My biggest issue was being able to verify the correctness of the core results (i.e., Theorem 5.1). I find the reviewers' elaboration helpful and see the high-level idea better, though I still would not say I can rigorously validate all details (though this could be due to a lack of familiarity with the area). I have therefore raised my score.
> >
> > ## Result Presentation and Correctness
> >
> > I find the reviewers' elaboration here helpful and better understand the high-level idea. The training examples can be decomposed in a kind of kernel machine way, and Assumption 5.1 fairly directly allows us to conclude that the predictor will be positive if the ground truth is 1.
> >
> > However, the proof structure could be made much clearer. For example, consider the following sentences:
> >
> > > In particular, both the train and test NTK kernels assume a scaled identity that keeps different blocks from interfering with one another. Furthermore, since any test input activates at most one vector per block, the test diagonal elements of the test NTK kernel (measuring the similarity of the test input to each element of the training set) take only two possible values: one indicating “this block is active” and one indicating “this block is empty”.
> >
> > The equation for the scaled identity form appears after this second sentence, when it would be more natural to present it immediately after the first sentence, potentially on its own line.
> >
> > Similarly, on line 219, this equation could be given its own line.
> >
> > On line 225, I would suggest suggest adding a paragraph break when you transition to proving the two directions.
> >
> > ## Other Comments
> >
> > > Our goal was not to learn a universal computer end-to-end.
> >
> > I still think it is important to highlight that you are learning one instruction rather than a larger function in the Turing-complete class. I see some potential for readers to be confused, e.g., assume your results apply to any learning any function representable by a composition of instructions, when in fact your result applies for learning individual instructions.
> >
> > Thank you for your answers to questions 1 and 2 as well.

---

> > > ### Author Response · Authors · 2025-08-04
> > >
> > > We sincerely thank the reviewer for meaningfully participating in the discussion period and for raising their score.
> > >
> > > Regarding the presentation of our main results, we find the reviewer’s suggestions about clarity valuable and plan to incorporate them into the camera-ready version. As a note, the equations were originally formatted as suggested, but were inlined for the submission due to the nine-page limit. With the additional page available for the camera-ready version, we will restore the more natural formatting to further improve readability.
> > >
> > > About the reviewer’s comment on instruction learning, we agree that this distinction is important, and we commit to explicitly stating that as well as revising relevant parts of the manuscript.
> > >
> > > We thank the reviewer once more for their insightful comments.

---

### Decision · Program_Chairs · 2025-09-17

**Decision:**

Accept (poster)

**Comment:**

The paper presents a novel theoretical result showing that an ensemble of two-layer networks trained in the infinite-width limit can learn to compute four fundamental binary functions. The analysis relies on reduction of these tasks to local pattern matching and analysis of learnability in the NTK regime.

The reviewers appreciated the rigorous theory provided in the paper, as well as the NTK-analysis and the importance of the tasks studied in the paper. During the review process, some reviewers were concerned about clarity of the assumptions and the results. Reviewers also raised concerns on comparison to prior work. However, after discussion with the authors it seems that all reviewers were satisfied with the further clarification provided by the authors and agree that this paper should be accepted. I therefore decided to accept this paper.